# A Reproduction Study of Weight-Based Mechanistic Interpretability in Bilinear MLPs

## Abstract

Mechanistic interpretability typically explains a trained network by analyzing its activations, at the cost of training auxiliary models such as sparse autoencoders (SAEs). Pearce et al. (2025) pursue an alternative: bilinear MLPs, whose omission of element-wise nonlinearities makes each layer an exact quadratic form, so interpretable features can be read directly from the weights via eigendecomposition of the layer's interaction tensor. We present a systematic reproduction of that work. The vision claims reproduce fully: regularization induces low-rank, visually interpretable eigenstructure, and our ablations sharpen the original account by identifying weight decay—rather than noise augmentation—as the dominant cause. The language claims reproduce only partially: we confirm the qualitative discovery of sentiment-negation circuits, but the reported prevalence of low-rank feature interactions holds for only one of three publicly released models (a second becomes consistent once rarely-active dictionary features are excluded). We identify two candidate factors that the original paper leaves unreported—SAE training duration and the underlying model's training compute—and a third that is structural: the public unavailability of matching SAE artifacts, which makes part of the original configuration impossible to replicate exactly. Beyond reproduction, we test whether weight-based features are genuinely structural rather than dataset-specific: regularized bilinear MLPs transfer across handwritten-digit datasets and recognize geometrically similar letters, and we introduce Quadratic Form Similarity, a weight-space metric that separates structurally similar from dissimilar class pairs where eigenvector cosine similarity cannot. Finally, we show on MNIST-scale models that the low-rank structure the original paper discovers post-hoc can instead be enforced during training via Canonical Polyadic (CP) decomposition: at matched full training, CP models match dense accuracy and effective rank, and a bridge experiment connecting the two extensions shows their decision surfaces substantially coincide with the dense ones—enforced and discovered structure converge. Our results are consistent with weight-based interpretability as a viable paradigm, while demonstrating that its reproducibility hinges on artifact availability and training-compute details that interpretability research rarely reports.

## 1 Introduction

Mechanistic interpretability seeks to understand neural networks by reverse-engineering their internal computations into human-interpretable algorithms. The dominant approach trains Sparse Autoencoders (SAEs) post-hoc on model activations to discover interpretable features (Bricken et al., 2023; Cunningham et al., 2023). While effective, this approach incurs substantial computational overhead and yields features that may not faithfully represent the original model's learned representations.

Pearce et al. (2025) pursue an alternative paradigm: *intrinsic* interpretability through architectural design. Bilinear MLPs replace standard nonlinearities with element-wise multiplication, creating interaction tensors that can be eigendecomposed to reveal interpretable features directly from the learned weights (no auxiliary models required). The paper demonstrates this on vision (MNIST classification) and language (sentiment negation circuits in transformers).

The paradigm has a longer lineage: Concept Bottleneck Models route predictions through human-specified concepts (Koh et al., 2020), and GAMI-Net restricts the function class to additive main effects and sparse pairwise interactions (Yang et al., 2021), imposing interpretability at the *representation* level. Bilinear layers impose it at the *computation* level: Sharkey (2023) first argued that a Gated Linear Unit (GLU) without element-wise nonlinearity is expressible through a third-order tensor and therefore analyzable with linear-algebraic tools, and Pearce et al. (2025) is the empirical realization of that proposal. If learned eigenvectors correspond to human-interpretable concepts, the model's decision process becomes auditable from its parameters alone. Whether this promise survives independent reproduction—and which unstated experimental details it depends on—is the question this report answers.

We reproduce both main experimental sections of Pearce et al. (2025):

- **Vision Reproduction (Section 4.1):** Fully reproduced. Regularization induces low-rank interaction tensors whose top eigenvectors form interpretable digit templates, at negligible accuracy cost; our ablation disentangles the two regularizers, showing that weight decay—not noise augmentation—is the primary driver of low-rank structure.
- **Language Reproduction (Section 4.2):** Partially reproduced. We confirm AND-gate negation circuits—output features requiring the conjunction of two input feature clusters—with semantically contrasting features (not-bad vs. not-good), on three released bilinear transformers named by training corpus (ts-* = TinyStories, fw-* = FineWeb-EDU; Section 3.4). However, under the original >0.75 rank-2 correlation threshold (agreement between a feature's activation and its reconstruction from the top two eigenvectors, Section 3.3), the claimed prevalence of low-rank interactions holds only for fw-small (65%), not ts-medium (32%) or fw-medium (45%; sensitivity across thresholds in Table 3). We trace the gap to two unreported factors—SAE training duration and model training compute—and to SAE checkpoints that are not all publicly available (Section 2.3); the language results therefore constitute a constrained replication under public artifacts rather than an exact reproduction.
- **Extension 1 (Section 5):** We test whether the learned features are genuinely structural: regularized bilinear MLPs transfer across MNIST, EMNIST, and USPS, and geometrically similar letters are classified as their digit counterparts. We propose Quadratic Form Similarity (QFS) to quantify structural alignment between decision surfaces where eigenvector cosine similarity fails, and compare it against Centered Kernel Alignment (CKA) as an established baseline.
- **Extension 2 (Section 6):** We explore Canonical Polyadic (CP) decomposition as an architectural constraint that enforces during training the low-rank structure the original paper finds post-hoc, and use QFS to bridge the two extensions: at matched full training, CP models match dense accuracy (97.8% vs. 97.5%) and effective rank, and their decision surfaces substantially coincide with the dense ones (Section 6.4)—the accuracy and surface gaps seen at shorter budgets are training-regime effects. Both extensions are deliberately scoped as MNIST-scale proof-of-concepts; we state their limits in Section 7.5.

## 2  Scope of Reproducibility

We reproduce the main experimental sections of Pearce et al. (2025): their Section 4 (Vision), interpretable eigendecomposition on MNIST and Fashion-MNIST, and Section 5 (Language), negation-circuit discovery in bilinear transformers via Sparse Autoencoder (SAE) analysis.

### 2.1  Claims to Verify

**Section 4: Vision (Image Classification).**

V1 *Low-rank emergence*: regularization induces low-rank interaction tensors.
V2 *Interpretable eigenvectors*: top eigenvectors of regularized models form digit-like templates rather than noise.
V3 *Overfitting without regularization*: unregularized models yield high-rank, noise-like eigenvectors.
V4 *Regularization roles*: noise augmentation and weight decay play distinct, complementary roles in producing V1–V3.

V5 *Cross-seed consistency and truncation*: top eigenvectors are stable across seeds, and truncating to a few eigenvectors per class preserves accuracy.

V6 *Adversarial generation*: eigenvectors can be used constructively, e.g. to build adversarial masks stronger than random ones.

**Section 5: Language (Negation Circuits).**

L1 *SAE feature discovery*: SAEs on bilinear transformers yield interpretable sparse features.

L2 *Negation/AND-gate circuits*: some features implement AND-gate circuits with opposing eigenvalues for "not + positive" vs. "not + negative".

L3 *Low-rank interactions*: most SAE features admit high rank-2 correlation.

L4 *Sparse interactions*: feature interaction matrices are sparse and block-structured.

L5 *Cross-model generalization*: L3–L4 hold across model sizes.

## 2.2 Success Criteria

These criteria were fixed before running the reproduction experiments. (Metrics are defined in Sections 3: effective rank in Section 3.2, rank-2 correlation and SAE expansion factors in Section 3.3.)

**Vision:** Effective rank ratio (regularized/unregularized) $< 0.5$; top eigenvectors visually resemble digit templates; test accuracy within 2% of paper ($\sim$95–98%); results consistent across 5 seeds.

**Language:** AND-gate circuits show opposing eigenvalue signs; more than 50% of included features—the minimal reading of the paper's "most features"—show $>0.75$ rank-2 correlation, with sensitivity across $>0.40$–$>0.75$ and bootstrap confidence intervals reported in Table 3; interaction matrices show interpretable block structure.

## 2.3 Deviations from the Original Setup

Our language setup necessarily deviates from Pearce et al. (2025) in three ways, which bound how directly our numbers can be compared to theirs:

- **SAE expansion factor.** The paper's configuration uses 4× expansion SAEs. For `fw-medium` at the analyzed layer, only 8× expansion SAEs are publicly released, so our `fw-medium` results use 8× expansion (8,192 rather than 4,096 dictionary features), which affects feature granularity and sparsity. (The released SAE repositories have been unchanged since November 2024, so this availability gap is stable rather than transient.)

- **Analyzed layers.** SAE checkpoint availability differs per model, so the correlation analysis uses different layers for different models rather than a single common layer.

- **SAE training duration.** The paper does not report how long its SAEs were trained; public checkpoints may not match that duration, and we show (Section 4.2) that this variable alone changes rank-2 correlation by a factor of 2.8.

Vision experiments involve no such deviations: we retrain all models from scratch with the paper's published hyperparameters. Consequently, we interpret our vision results as a reproduction and our language results as a *constrained replication* under publicly available artifacts.

## 2.4 Results Summary

Table 1 summarizes our reproduction outcomes: all six vision claims are fully reproduced, while language claims are partially reproduced—AND-gate circuits are confirmed, but low-rank correlation under the original $>0.75$ threshold is reproduced only for fw-small (Section 4.2).

Table 1: Summary of reproduction results. ✓ = reproduced, × = not reproduced, ∼ = partially reproduced under substituted artifacts (the constrained-replication setting of Section 2.3).

| Claim | Description | Status | Claim | Description | Status |
|-------|-------------|--------|-------|-------------|--------|
| | *Vision (Section 4)* | | | *Language (Section 5)* | |
| V1 | Low-rank emergence | ✓ | L1 | SAE feature discovery | ✓ |
| V2 | Interpretable eigenvectors | ✓ | L2 | Negation/AND-gate circuits | ✓ |
| V3 | Overfitting without reg. | ✓ | L3 | Low-rank interactions | ∼ |
| V4 | Regularization roles | ✓ | L4 | Sparse interactions | ✓ |
| V5 | Cross-seed consistency and truncation | ✓ | L5 | Cross-model generalization | ∼ |
| V6 | Adversarial generation | ✓ | | | |

## 3 Methodology

### 3.1 Bilinear Layer Architecture

The bilinear layer is a member of the Gated Linear Unit family with the element-wise nonlinearity removed: instead of applying a fixed activation function, it multiplies two linear projections of the input element-wise,

$$\mathbf{y} = (\mathbf{W}_l \mathbf{x}) \odot (\mathbf{W}_r \mathbf{x}) \tag{1}$$

where $\mathbf{W}_l, \mathbf{W}_r \in \mathbb{R}^{d_{\text{hidden}} \times d_{\text{in}}}$ are learnable weight matrices and $\odot$ denotes the Hadamard product. The layer is still nonlinear in $\mathbf{x}$ (the left projection gates the right), so expressivity remains competitive with standard activations (Pearce et al., 2025); what changes is analyzability. A ReLU or GELU MLP has no closed algebraic form—which linear region applies depends on the input—so its weights cannot be interpreted without running data through the network. A bilinear layer, by contrast, is *exactly* quadratic in its input: every output coordinate is a quadratic form, expressible through a third-order interaction tensor

$$y_c = \sum_{i,j} B_{cij} x_i x_j \tag{2}$$

where $B_{cij} = \sum_h W_{\text{head},ch}(W_l)_{hi}(W_r)_{hj}$ and $W_{\text{head}} \in \mathbb{R}^{d_{\text{out}} \times d_{\text{hidden}}}$ is the output projection. This closed form is the foundation of the approach (Sharkey, 2023): $B$ is a complete, data-independent description of the layer's computation, so linear-algebraic tools applied to $B$ characterize the model's behavior on *all* inputs, not just a sampled dataset.

### 3.2 Eigendecomposition for Interpretability

For each output class $c$, we extract the $d_{\text{in}} \times d_{\text{in}}$ interaction matrix $B_c$ and symmetrize it, $B_c^{\text{sym}} = \frac{1}{2}(B_c + B_c^\top)$, which removes arbitrary asymmetry from the $W_l/W_r$ factorization while preserving the quadratic form $\mathbf{x}^\top B_c \mathbf{x}$. The symmetric matrix admits eigendecomposition:

$$B_c^{\text{sym}} = \sum_{r=1}^{d_{\text{in}}} \lambda_r^{(c)} \mathbf{v}_r^{(c)} (\mathbf{v}_r^{(c)})^\top \tag{3}$$

Substituting into eq. (2), the class-$c$ logit decomposes as

$$y_c(\mathbf{x}) = \sum_r \lambda_r^{(c)} \left( \mathbf{v}_r^{(c)\top} \mathbf{x} \right)^2, \tag{4}$$

i.e., each eigenvector $\mathbf{v}_r^{(c)} \in \mathbb{R}^{d_{\text{in}}}$ is a template whose squared alignment with the input contributes evidence *for* class $c$ if $\lambda_r^{(c)} > 0$ and *against* it if $\lambda_r^{(c)} < 0$. For image inputs each eigenvector can be rendered directly as an image; if the eigenspectrum is *low-rank* (few dominant eigenvalues), the class decision depends on only a few such templates.

We quantify spectral concentration using effective rank (Roy & Vetterli, 2007):

$$\text{EffRank}(\boldsymbol{\lambda}) = \left( \frac{\|\boldsymbol{\lambda}\|_1}{\|\boldsymbol{\lambda}\|_2} \right)^2 = \frac{(\sum_i |\lambda_i|)^2}{\sum_i \lambda_i^2} \tag{5}$$

This scale-invariant measure ranges from 1 (single dominant eigenvalue) to $d$ (uniform spectrum).

### 3.3 Language Model Analysis with SAEs

In transformers, individual neurons are rarely interpretable because models store more concepts than they have dimensions, superimposing multiple concepts on each direction (Elhage et al., 2022). Sparse Autoencoders (SAEs) (Bricken et al., 2023; Cunningham et al., 2023) address this by re-representing an activation vector $\mathbf{h}$ in an overcomplete dictionary, $\mathbf{z} = \mathrm{TopK}(\mathbf{W}_{\mathrm{enc}}\mathbf{h})$, where the dictionary is 4–8× wider than $\mathbf{h}$ (the *expansion factor*) and TopK keeps the $k$ largest coefficients per token. Trained to reconstruct activations under this sparsity constraint, SAE features tend to be *monosemantic*: each corresponds to a single human-recognizable concept (e.g., negation, a sentiment).

The language experiments combine SAEs with bilinear decomposition in two ways. For the *correlation analysis* (paper Figure 9), an SAE at the MLP output defines sparse output features: by the bilinear structure, the pre-activation of output feature $f$ is a quadratic form in the raw MLP input $\mathbf{x}$ (residual-stream basis), $z_f^{\mathrm{out}} \approx \mathbf{x}^\top Q_f \mathbf{x}$, where $Q_f$ is the layer's interaction tensor projected onto feature $f$'s SAE encoder direction; eigendecomposing $Q_f$ reveals the dominant input directions driving $f$. For the *circuit case study* (paper Figure 8), an SAE at the MLP input additionally re-expresses those input directions in a sparse feature basis, so interactions can be read as feature-to-feature circuits. The original paper's quantitative claim is that for "most features" a rank-2 truncation of $Q_f$ suffices: reconstructing $z_f^{\mathrm{out}}$ from only the top two eigenvectors correlates >0.75 with the true activation—i.e., the feature is implemented by an interaction of essentially two input directions (the original does not specify the correlation variant; we use Pearson over each feature's active tokens). We use the authors' pretrained SAEs with TopK sparsity ($k = 30$).

### 3.4 Datasets

**Vision Datasets.** We use MNIST (LeCun et al., 1998) (60K train / 10K test, 28×28 grayscale handwritten digits) and Fashion-MNIST (Xiao et al., 2017) (60K train / 10K test, 28×28 grayscale clothing images). Images are normalized to $[0, 1]$ and flattened to 784-dimensional vectors. For cross-dataset experiments (Section 5), we additionally use EMNIST-Digits/Letters (Cohen et al., 2017) and USPS (Hull, 1994) (upscaled from 16×16 to 28×28).

**Language Datasets.** The pretrained bilinear transformers use two datasets: **TinyStories** (Eldan & Li, 2023) (2.1M synthetic children's stories, ∼470M tokens) for `ts-medium`, and **FineWeb-EDU** (Penedo et al., 2024) (educational web text subset of FineWeb, 1.3T tokens) for `fw-small` and `fw-medium`. For correlation analysis (Figure 3), all three models use ∼1.57M tokens (1,572,864) from their respective validation sets.

### 3.5 Experimental Setup

For vision experiments, we train bilinear MLPs on MNIST and Fashion-MNIST under four regularization configurations (Table 2), each with 5 random seeds (42–46). Architecture: $d_{\mathrm{hidden}} = 256$, 100 epochs, batch size 2048, AdamW optimizer with lr $10^{-3}$ and cosine annealing. For language experiments, we analyze pretrained bilinear transformers (Table 4, Appendix C). Due to SAE availability constraints (Section 2.3), our primary model is `fw-medium` rather than the paper's `ts-medium`. We implemented a memory-efficient streaming algorithm for eigendecomposition to handle large interaction tensors.

Table 2: Regularization configurations (from paper Appendix G.1). Noise augmentation adds i.i.d. Gaussian noise of the stated standard deviation to input pixels during training.

| Configuration | Noise Std | Weight Decay |
|---|---|---|
| `none` | 0.0 | 0.0 |
| `noise` | 0.5 | 0.0 |
| `wd` | 0.0 | 1.0 |
| `full` | 0.5 | 1.0 |

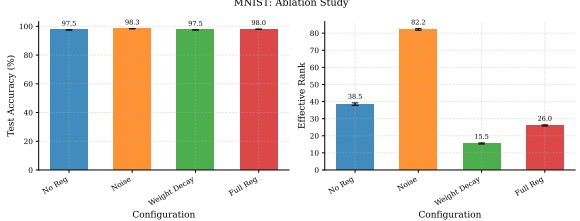
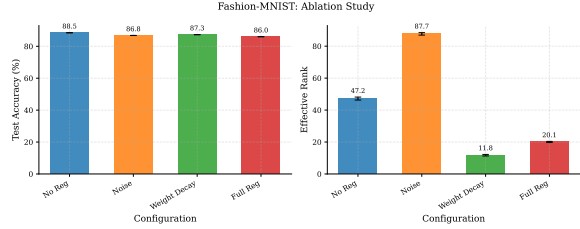

(a) MNIST: Accuracy vs. effective rank

(b) Fashion-MNIST: Accuracy vs. effective rank

Figure 1: Ablation study showing accuracy-interpretability trade-off across regularization configurations. Weight decay achieves the lowest effective rank on both datasets while maintaining competitive accuracy. The effective rank ratio (regularized/unregularized) meets the paper's $< 0.5$ criterion.

## 4 Reproduction Results

### 4.1 Vision Results (Paper Section 4)

#### 4.1.1 Main Result: Regularization Induces Low-Rank Structure (Claim V1)

Figure 1 confirms the paper's central claim: regularization induces low-rank structure in the bilinear interaction tensor. On MNIST, weight decay is the primary driver, achieving the lowest effective rank (15.5, vs. 26.0 for full regularization and 38.5 unregularized), while noise augmentation increases effective rank but improves accuracy. The effective rank ratio (regularized/unregularized, $15.5/38.5 \approx 0.40$) meets the paper's $< 0.5$ criterion. Eigenvalue distributions reveal sharp spectral decay with regularization (Figure 7, Appendix B), concentrating variance in 10–20 eigenvectors. Top eigenvectors are interpretable digit-like patterns under regularization, while unregularized models show diffuse, noise-like patterns (Figures 16 and 17).

#### 4.1.2 Additional Vision Results (Claims V5, V6)

We confirm the paper's claim that bilinear MLPs maintain near-full accuracy with low-rank truncation: truncating to around 20 eigenvectors per class is sufficient even as model size increases, and top eigenvectors are highly consistent across random seeds, especially for larger models (Figure 8, Appendix B). We also reproduce the original paper's Figures 4, 6, and 7; full visualizations are in Appendix B. Increasing noise improves eigenvector interpretability while trading off with effective rank (Figure 9). On the paper's challenge task—a binary classifier whose label is positive iff an image's pixel-space cosine similarity to a fixed target-digit image (or its inverse) exceeds a threshold—the top eigenvector closely resembles the target digit (Figure 11), and eigenvector-derived masks cause significantly larger accuracy drops than random masks (Figure 13), confirming V6.

### 4.2 Language Results (Paper Section 5)

We reproduce negation circuit discovery using pretrained bilinear transformers and SAEs from HuggingFace.[1] Our protocol: pass tokenized batches through each model, record raw `mlp_in` activations $\mathbf{x}$ and `mlp_out` activations, encode the latter with the `mlp-out` SAE to obtain true sparse feature activations, and compare them to the low-rank predictions $\sum_j \lambda_j (\mathbf{v}_j^\top \mathbf{x})^2$ from the eigenpairs of $Q_f$. We analyze all *active features* ($\geq 1$ active token) using Pearson correlation; features without a valid correlation (fewer than two active tokens, or degenerate variance such as constant activation) are excluded. **SAE availability gap: `ts-medium-scope`** (released for the model the paper calls `ts-tiny`) lacks `mlp-in` SAEs for layer 4, so the paper's Figure 8 setup cannot be reproduced; we use `fw-medium` (layer 7, expansion 8), which has both SAE types.

#### 4.2.1 Negation Circuit Discovery (Paper Figure 8)

For output feature $f$, the interaction matrix $Q_f$ captures how input feature pairs contribute to $f$'s activation: $z_f^{\text{out}} \approx (\mathbf{z}^{\text{in}})^\top Q_f \mathbf{z}^{\text{in}}$. Figure 2 shows this for negation features in fw-medium: the interaction submatrix over

---

[1]SAE repositories: `tdooms/fw-medium-scope`, `tdooms/ts-medium-scope`, `tdooms/fw-small-scope`; the `-scope` suffix denotes the SAE release accompanying each model, with SAEs trained at the MLP input (`mlp-in`) and output (`mlp-out`) of each layer.

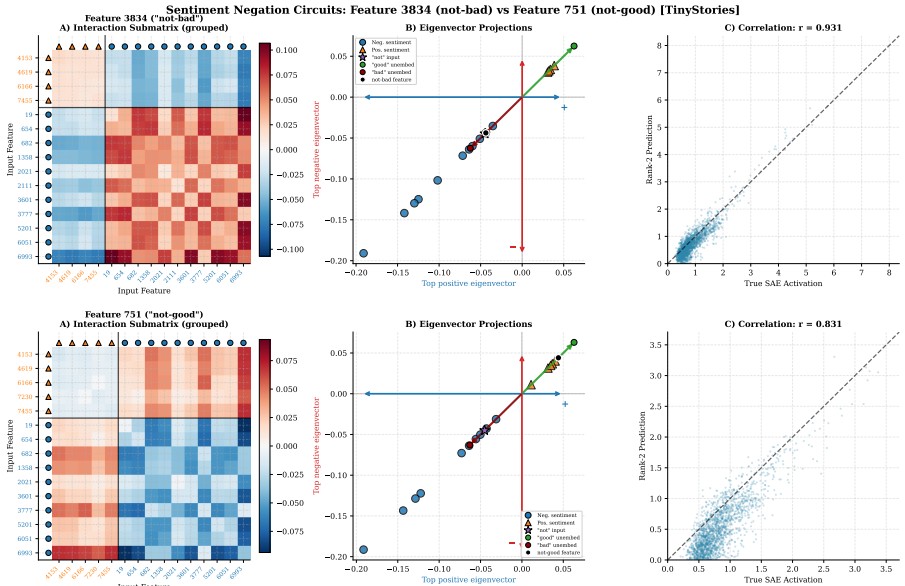

Figure 2: Negation circuit discovery (fw-medium, layer 7, TinyStories evaluation). Top: feature 3834 ("not-bad", $r = 0.93$); bottom: feature 751 ("not-good", $r = 0.83$). (A) Interaction submatrix $Q_f$ shows block structure: positive/negative sentiment features interact oppositely. (B) Eigenvector projections reveal clustering: sentiment features separate along the "good"↔"bad" axis.

the top-15 input features has block structure (positive/negative sentiment features interact oppositely), and eigenvector projections cluster sentiment features along the "good"↔"bad" axis, with the output feature at the intersection of negation and sentiment. This reveals the AND-gate circuit: *negation* AND *sentiment* → *negated-sentiment*. This case study is a qualitative existence result: because features are selected for strong interaction structure, it cannot speak to prevalence (that is the role of the threshold analysis below), and note that it uses layer 7 while the prevalence analysis uses layer 10 of the same model, so the L2 and L3 verdicts come from different depths.

**Feature selection and interpretation:** Following the methodology of the original authors' language tutorial[2], we select output features by ranking all 8,192 features by their top eigenvalue magnitude—a heuristic for strong interaction structure. Features 3834 and 751 emerge as the top two, with strongly negative cosine similarity ($-0.73$) between their SAE decoder directions—consistent with the tutorial's observation that the two features span a "somewhat linear subspace." Top-activating tokens give the semantics: feature 3834 fires on negated negative words ("no losses", "not lost"), feature 751 on negated positive words ("not harmless", "doesn't go away")—semantically contrasting negation features (not-bad vs. not-good) with strongly anti-aligned decoder directions ($-0.73$; the analogous pair the paper reports in ts-medium is more extreme, at $-0.975$). On FineWeb-16k (`tdooms/fineweb-16k`, a tokenized slice of the model's FineWeb-EDU training distribution released by the original authors), the circuit's correlations remain strong (0.91/0.82 vs. 0.93/0.83 on TinyStories), demonstrating robustness to distribution shift (Appendix D).

### 4.2.2 Low-Rank Correlation Analysis (Paper Figure 9)

The paper claims most features achieve >0.75 rank-2 correlation; because "most" is not formally specified, we report results across thresholds from >0.40 to >0.75 (Table 3). This analysis uses fw-medium layer 10 rather than the circuit case study's layer 7: layer 10 lies at 2/3 network depth, where the paper's Figure 9 configurations are specified, while layer 7 matches the tutorial's configuration (features 3834/751). For each token, we record the raw MLP input **x** and the MLP output, encoding the latter with the `mlp-out` SAE ($k$=30 TopK sparsity, matching the paper) to obtain the true sparse feature activations. Only the output-side SAE is required (the input side of $Q_f$ lives in the raw residual-stream basis), which is why the

---

[2] `tutorials/2_language.ipynb` in the official code release of Pearce et al. (2025) (`https://github.com/tdooms/bilinear-decomposition`), which demonstrates the negation-circuit analysis on fw-medium features 3834/751; hereafter "the tutorial."

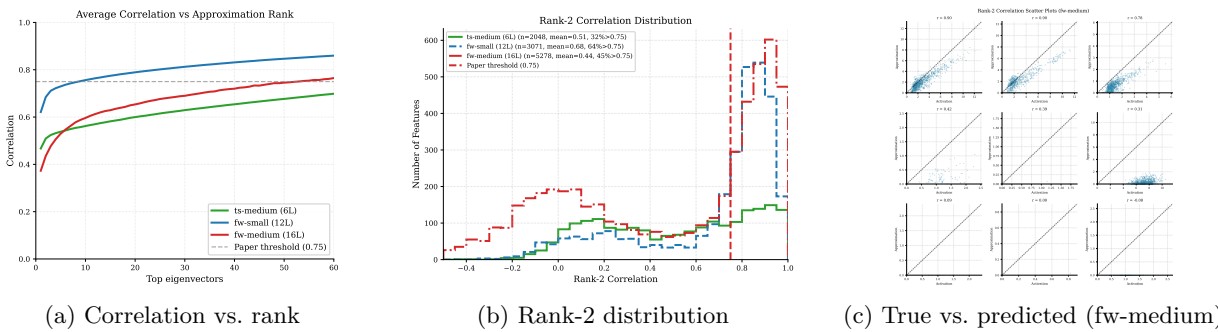

(a) Correlation vs. rank        (b) Rank-2 distribution        (c) True vs. predicted (fw-medium)

Figure 3: Low-rank correlation (Paper Figure 9). ts-medium (L4, exp=4, $n$=2048), fw-small (L8, exp=4, $n$=3071), fw-medium (L10, exp=8, $n$=5278). $n$ = features with a valid Pearson $r$ (at least two active tokens, non-degenerate variance). Panel (c) shows features deliberately stratified across the correlation range (three each with high, medium, and low rank-2 correlation), not a random sample.

analysis can proceed for ts-medium despite the missing `mlp-in` SAEs noted above. For each feature $f$ with a valid Pearson correlation ($n$ denotes the number of such features), we eigendecompose its interaction matrix $Q_f$ and compute the correlation between true and low-rank predicted activations across the feature's active tokens (token counts and datasets in Section 3.4).

### 4.2.3 Rank-2 Correlation: Distribution and Sensitivity

The rank-2 correlation distributions (Figure 3b) tell a richer story than any single threshold: fw-medium's median (0.649) sits above ts-medium's (0.550) despite a lower mean (0.435 vs. 0.509) and lower >0.75 fraction (44.5% vs. 32.3%), reflecting a distribution concentrated around 0.5–0.7 but with fewer high-correlation features. Table 3 reports the fraction of features above several thresholds, with 95% bootstrap confidence intervals (10,000 resamples over features). The intervals are narrow ($\pm$1–2 pp): fw-small is separated from both other models at every threshold, and ts-medium and fw-medium are separated at the two strictest thresholds (including the original >0.75 criterion), so the headline verdict does not rest on feature-sampling noise; the two weaker models' intervals overlap at the looser thresholds (>0.40, >0.50). fw-small reproduces the "most features" claim robustly across all thresholds, while ts-medium and fw-medium fall below it and remain marginal even under weaker criteria.

Sensitivity to the *inclusion criterion* is larger still, and asymmetric across models. Our pipeline admits every feature with a valid correlation, including rarely-active ones, whose correlations are dominated by small-sample effects: empirically, features with $n_{\text{active}} \in [10, 200]$ have median $r \approx 0$, while features firing on only 2–3 tokens yield degenerate $|r| \approx 1$ (released in `min_active_sensitivity.json`). This affects only fw-medium: its larger 8$\times$ dictionary contains thousands of rarely-active features (minimum $n_{\text{active}} = 2$), whereas every ts-medium and fw-small feature is well-sampled (minimum $n_{\text{active}}$ of 1,399 and 146, respectively). Restricting to well-sampled features raises fw-medium's >0.75 fraction monotonically—44.5% (all valid, $n$=5,278) → 52.4% ($n_{\text{active}} \geq 50$, $n$=4,207) → 68.1% ($\geq 500$, $n$=2,942) → 73.4% ($\geq 1,000$, $n$=2,149)—while leaving the other two models' fractions unchanged. Under a well-sampled-features reading, fw-medium therefore *matches or exceeds* the original paper's 69%; ts-medium's shortfall (32.3%, computed entirely on well-sampled features) is robust to this choice. Because the original paper does not specify an inclusion criterion, we report both readings rather than adjudicate between them; for future reproductions we recommend pre-registering a minimum-activity criterion (e.g., $n_{\text{active}} \geq 1,000$) so the verdict is single-valued.

### 4.2.4 Root Cause Analysis

The paper notes correlation improves with SAE training time (Appendix H). We confirm this using fw-medium layer 12 checkpoints v0–v4 (Figure 18, Appendix C; 16$\times$ expansion): mean rank-2 correlation improves from 0.15 to 0.42 (2.8$\times$) with longer training, consistent with the paper's acknowledgment that "correlation drastically improves with longer SAE training."

**Model training duration hypothesis:** We observe a correlation between training compute (tokens/parameter, computed from the released model configurations and reported training-token counts) and

Table 3: Rank-2 correlation sensitivity across thresholds (features with valid Pearson $r$). Threshold columns give the percentage of features above each threshold, with 95% confidence intervals from a nonparametric bootstrap over features (10,000 resamples). The intervals quantify feature-sampling variability conditional on one model, one SAE checkpoint, and one token sample (run-level $n$=1 per row); they do not cover SAE-training uncertainty, which Section 7.3 shows is larger.

| Model | $n$ | Mean | Median | >0.40 (%) | >0.50 (%) | >0.60 (%) | >0.75 (%) |
|---|---|---|---|---|---|---|---|
| ts-medium | 2048 | 0.509 | 0.550 | 59.2 [57.1, 61.3] | 53.3 [51.2, 55.5] | 46.2 [44.1, 48.4] | 32.3 [30.3, 34.4] |
| fw-small | 3071 | 0.685 | 0.814 | 80.3 [78.9, 81.7] | 78.0 [76.5, 79.4] | 75.6 [74.0, 77.1] | 64.5 [62.8, 66.2] |
| fw-medium | 5278 | 0.435 | 0.649 | 57.0 [55.7, 58.3] | 54.4 [53.1, 55.7] | 51.7 [50.4, 53.1] | 44.5 [43.2, 45.9] |

low-rank behavior: ts-medium (83 tokens/param, 32%), fw-medium (95 tokens/param, 45%), and fw-small (198 tokens/param, 65%). Since more training tokens corresponds to longer training time, this parallels Figure 18: just as undertrained SAEs fail to capture low-rank structure, undertrained models (low tokens/param) may not develop the structured interactions that exhibit low-rank approximability. We caution that this correlation is itself criterion-dependent: it holds under the all-valid-features reading, but under the well-sampled reading above the ordering inverts for the FineWeb pair (fw-medium 73.4% vs. fw-small 64.5%), leaving only ts-medium's shortfall as evidence. We therefore regard the compute factor as the weaker of the two candidates—an instance of the criterion sensitivity Lesson 3 (Section 7.3) warns about, applied to our own hypothesis.

**SAE artifact limitation:** For fw-medium at 2/3 depth (layer 10–11), only 8× expansion SAEs are publicly available (4× exists only at layers 6 and 12; Section 2.3), preventing exact reproduction of the paper's 4× fw-medium configuration. The larger dictionary yields more sparse/specialized features that potentially lower average correlation (feature and exclusion counts in Appendix A).

## 5 Cross-Dataset Structural Robustness

Our two extensions operationalize one question from complementary directions: are the weight-space objects that eigendecomposition exposes—per-class quadratic decision surfaces—real structure, or artifacts of a particular training run? This section probes their *portability* across datasets; Section 6 probes their *enforceability* at training time; and a bridge experiment (Section 6.4) asks whether the discovered and the enforced objects coincide. Here we test whether bilinear MLPs learn reusable geometric primitives (e.g., "0-ness" as circularity) rather than idiosyncratic MNIST stroke patterns—a core promise of weight-based interpretability.

Throughout this section, **baseline** refers to models without regularization, while **regularized** refers to full regularization (noise std = 0.5, weight decay = 1.0) as in Table 2. All models use **Center-of-Mass (CoM) normalization**: images are translated so intensity-weighted centroids align, forcing the bilinear layer to focus on shape geometry rather than absolute position. We probe robustness under three distributional shifts: (1) **Writer independence**: MNIST ↔ EMNIST-Digits (same labels, different writers); (2) **Domain transfer**: MNIST → USPS (different resolution/collection); (3) **Geometric semantics**: MNIST → EMNIST-Letters, where letters are semantically different but geometrically similar (O→0, I→1, Z→2, S→5).

### 5.1 Functional Analysis: Regularization Enables Transfer

With the regularized CoM model, cross-dataset accuracy is near-perfect and symmetric for MNIST ↔ EMNIST-Digits (97.5% and 97.9%, Table 7, Appendix E), confirming that regularized bilinear features do not overfit to MNIST writers. For domain transfer to USPS, regularization yields +19 pp improvement (72.3% vs. 53.4%, Table 8, Appendix E). On geometrically similar letters (Table 9, Appendix E), the model's most confident prediction is exactly the corresponding digit: O→0 (99.6%), I→1 (87.1%), Z→2 (88.8%), S→5 (87.4%).

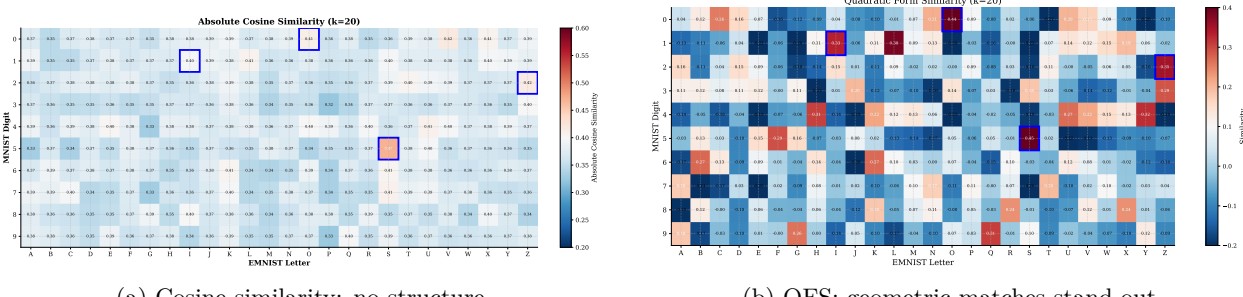

(a) Cosine similarity: no structure          (b) QFS: geometric matches stand out

Figure 4: **Cosine similarity vs. Quadratic Form Similarity**, MNIST digit classes against EMNIST letter classes. (a) Mean absolute cosine similarity between eigenvectors is nearly uniform—geometrically similar and dissimilar pairs are indistinguishable. (b) QFS ($k = 20$) exposes the shape-matched pairs (0–O, 1–I, 2–Z, 5–S) as bright entries against a near-zero background. The corresponding MNIST vs. EMNIST-Digits comparison is in Appendix E.

## 5.2 Structural Analysis: Quadratic Form Similarity

### 5.2.1 The Failure of Cosine Similarity

Direct eigenvector comparison via cosine similarity fails to distinguish geometrically similar pairs: over the balanced top-10 eigenvectors per class (the 5 largest-magnitude positive and 5 largest-magnitude negative eigenmodes; best-match absolute cosine, 5 seeds), MNIST-0 vs. EMNIST-O (similar) yields $0.312 \pm 0.016$ while MNIST-0 vs. EMNIST-X (dissimilar) yields $0.364 \pm 0.029$—the *dissimilar* pair actually scores slightly higher. Figure 4a shows this failure across all digit-letter pairs (per-eigenvector breakdown in Table 11, Appendix F). The root cause: bilinear eigensystems share eigenvector directions while class information resides in eigenvalue weights (Figure 20).

### 5.2.2 Quadratic Form Similarity

We propose comparing full decision surfaces via Quadratic Form Similarity:

$$\text{sim}(A, B) = \frac{\text{tr}(A \cdot B)}{\|A\|_F \|B\|_F} \in [-1, 1] \tag{6}$$

The formula itself is the standard matrix cosine, known as kernel–target alignment in kernel methods (Cristianini et al., 2001); our contribution is the object it is applied to—weight-derived, data-free interaction matrices of bilinear class logits—not the metric. QFS has a principled identity: writing $A_c := B_c^{\text{sym}}$ for the symmetrized interaction matrix of Section 3, the class-$c$ logit is the quadratic form $y_c(\mathbf{x}) = \mathbf{x}^\top A_c \mathbf{x}$ with Hessian $\nabla^2 y_c = 2A_c$ (bias and linear terms vanish at second order), so for full matrices QFS is exactly the Frobenius cosine between the Hessians of two class logits—a data-free, second-order functional similarity comparing the decision surfaces themselves rather than their responses on a probe set. It is invariant to orthogonal input transformations applied to both models and to positive rescaling, but deliberately *not* to independent rotations of each model's basis, which change the computed function (full properties in Appendix G). This is the same notion of similarity that CKA targets (Section 5.2.3), with the probe distribution removed. In practice our implementation approximates the full-matrix quantity from the top-$k$ eigenpairs of the interaction matrices ($k = 20$ by $|\lambda|$, as in the analyses below).[3] Incorporating both eigenvector alignment and eigenvalue magnitudes, QFS discriminates geometrically similar from dissimilar pairs: mean QFS over the four geometric pairs is 0.399 versus $-0.057$ over control pairs ($p < 10^{-4}$), with similar pairs converging to high similarity around $k = 20$ eigenvectors while dissimilar pairs remain near zero with non-overlapping 90% confidence intervals (Figure 22, Appendix E). Figure 4b shows the discrimination: bright entries at the shape-matched pairs (0–O, 1–I, 2–Z, 5–S), and a strong diagonal for MNIST vs. EMNIST-Digits (Figure 21,

---

[3]On this section's headline pair (CoM-normalized MNIST vs. regularized EMNIST-Letters, seed 42) the eigenpair estimator agrees with the exact Frobenius cosine of rank-20 truncated input-space interaction matrices to 0.023 (max) and 0.007 (mean) over the full $10 \times 26$ class grid; the estimator is exact when the eigenvectors are orthonormal, as in the input-space recomputation of Section 6.4.

Appendix E). Table 10 (Appendix E) confirms specificity: 3/4 pairs rank #1, with mean rank 1.2 vs. random baseline of 13.

### 5.2.3 Comparison with CKA

We benchmark QFS against linear CKA (Kornblith et al., 2019) on the same class-pair discrimination task, instantiated two ways: on rank-$k$ spectral representations $R_c = [\sqrt{|\lambda_i|}\,(\mathbf{v}_i^\top \mathbf{x})]_{i \le k}$ ($k$=20, matching the QFS truncation) and on the scalar quadratic outputs $y_c(\mathbf{x}) = \mathbf{x}^\top A_c^{(k)} \mathbf{x}$ (rank-$k$ truncated, preserving eigenvalue signs), both over the union of the MNIST and EMNIST-Letters test sets (Appendix H, Table 12). Rank-$k$ CKA fails to separate geometric from mismatched pairs (0–X scores 0.81 vs. 0–O's 0.80): its real-valued feature map discards eigenvalue *signs*, precisely the information that distinguishes bilinear class circuits. Scalar CKA, the closest data-dependent analogue of QFS, is genuinely competitive—it even ranks all four expected pairs first, where QFS ranks 3/4 first—but its similar and control distributions *overlap* in all five seeds (0–X $\approx$ 0.50 exceeds 2–Z $\approx$ 0.29), whereas QFS achieves complete separation (minimum similar > maximum control) in every seed, with pooled means over similar vs. control pairs of $0.399 \pm 0.006$ vs. $-0.057 \pm 0.003$ (identical two-sample $t$-tests over the 12 pairs: $p < 10^{-4}$ for QFS in every seed, $p \approx 2 \times 10^{-3}$ for scalar CKA; given only 12 non-independent pairs, we regard the separation property as the more meaningful statistic). QFS is also computed from weights alone—it characterizes the decision surface globally, without an evaluation distribution—whereas CKA requires probe data and inherits sensitivity to its choice and composition (Ding et al., 2021; Davari et al., 2023).

### 5.3 Conclusion

Regularized bilinear MLPs learn **structural features that transfer across datasets**: the "0" circuit is highly similar to the "O" circuit across independently trained models (mean QFS over geometric pairs $\approx$ 0.40 vs. $\approx$ −0.06 over control pairs, $p < 10^{-4}$). This strengthens the transparency claim—eigenvector features represent genuinely reusable shapes, not dataset-specific artifacts.

## 6 CP-Decomposition Analysis

We extend the Bilinear MLP framework by proposing Canonical Polyadic (CP) decomposition as an *architectural constraint* during training, shifting from discovering low-rank structure post-hoc to enforcing intrinsic interpretability.

### 6.1 Motivation and Methodology

**Why CP?** Among tensor decompositions (Kolda & Bader, 2009), CP is the natural choice for *interpretability* for two reasons. First, it is the direct generalization of what Pearce et al. (2025) already do: their per-class eigendecomposition writes each interaction matrix as a sum of symmetric rank-1 terms, and CP extends exactly this additive rank-1 structure to the full third-order tensor—each term reading "input directions $\mathbf{b}_r$ and $\mathbf{c}_r$ jointly excite output direction $\mathbf{a}_r$." Second, CP is essentially unique: under Kruskal's condition, its rank-1 factors are identified by the tensor itself up to permutation and scaling (Kruskal, 1977). The main alternative, Tucker decomposition, factors the tensor into a dense core and per-mode bases that can be counter-rotated against the core without changing the tensor, so any per-component "feature" is an arbitrary basis choice; Tucker is preferable for pure compression, but for explanation, identifiable rank-1 units are the property we need. Appendix K verifies both properties empirically on our trained interaction tensor: rotated Tucker factorizations reproduce the tensor to machine precision yet yield visually unrelated per-component "features," whereas CP factors recovered from independent random initializations largely agree up to permutation and sign. Unlike eigendecomposition, CP also does not enforce orthogonality, allowing factors to overlap naturally rather than being forced apart into entangled templates.

CP-decomposition parameterizes the interaction tensor of eq. (2) directly as a sum of rank-1 tensors:

$$B_{cij} = \sum_{r=1}^{R} \lambda_r\, a_{cr}\, b_{ir}\, c_{jr} \qquad (7)$$

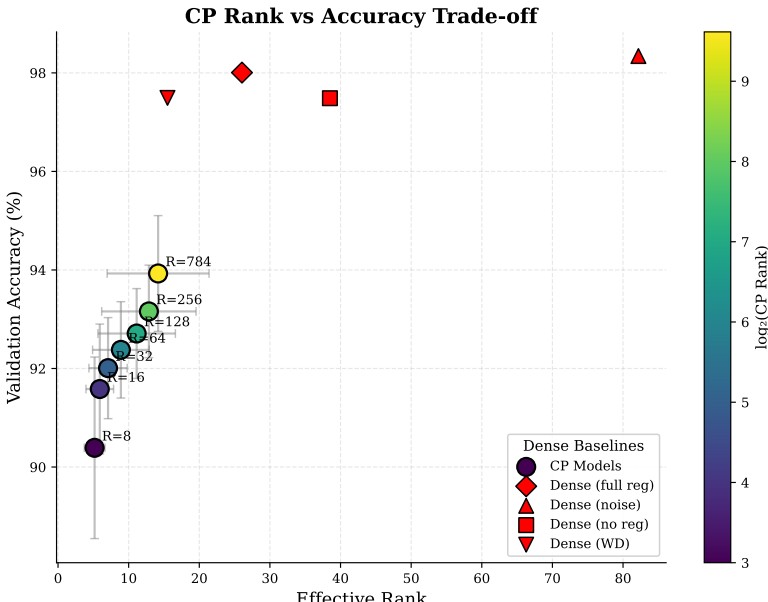

Figure 5: **Accuracy-Interpretability Pareto Frontier.** Fixed CP forces extreme sparsity (eff. rank ∼3.5) at accuracy cost. Lambda and Gated CP maintain competitive accuracy while reducing effective rank. Protocol note: the dense baselines shown are mini-batch trained, while all CP points are full-batch trained (100 optimizer steps); a budget-matched dense control sits at (eff. rank 6.1, 92.4%), so the apparent accuracy gap to the dense baselines is a training-budget effect (Section 6.4).

where $R$ is the maximum rank, $\mathbf{a}_r, \mathbf{b}_r, \mathbf{c}_r$ are learnable factor vectors (output-mode and two input-mode directions), and $\lambda_r$ are learnable scaling factors (Veeramacheneni et al., 2022). We investigate three variants, distinguished by how the component scales are parameterized: **Fixed CP:** factors trained directly under a hard rank constraint—a strict complexity bound. **Lambda CP:** learnable per-component scales $\lambda_r$ on column-normalized factors (Veeramacheneni et al., 2022). **Gated CP:** sigmoid gates on components, an $L_0$-style parameterization (Cao et al., 2024). The architecture supports explicit $L_1/L_0$ sparsity penalties on the scales and gates, but our runs disable them (coefficients 0): the low effective ranks reported below therefore emerge from the scale parameterizations combined with weight decay alone. All variants trained on MNIST with weight decay 0.1 (lighter than the dense `wd` config's 1.0; Section 6.4 brackets this asymmetry) across rank sweep $R \in \{32, 64, 128, 256\}$.

## 6.2 Quantitative Analysis: Accuracy-Interpretability Trade-off

Figure 5 presents the Pareto frontier; Table 13 (Appendix I) and the complete rank sweep in Table 14 (Appendix J) provide the numbers. Fixed CP operates as a strict bottleneck with eff. rank ∼3.5 regardless of allocated capacity, at ∼6% accuracy cost (91.9% vs. 97.5%). Lambda/Gated CP at $R = 256$ achieve 93.8% accuracy with eff. rank 17.5—comparable to dense models with weight decay (eff. rank 15.5)—while yielding qualitatively more localized factors in the visual analysis below. (Accuracy comparisons here are across protocols—full-batch CP vs. mini-batch dense; Section 6.4 shows that under the dense protocol CP reaches dense-level accuracy, $97.81 \pm 0.07\%$.)

## 6.3 Qualitative Analysis: Disentangled Factors

Figure 23 (Appendix J) compares eigenvectors from Dense vs. CP models. Dense eigenvectors resemble superimposed digit templates—entangled representations combining strokes for orthogonality. CP factors appear qualitatively more **localized** and parts-like: top loops, vertical strokes, bottom curves that can be linearly combined. This localization is not an artifact of CP's shorter training protocol: the budget-matched dense control of Section 6.4 yields diffuse, noisy eigenvectors rather than parts (Figure 24, Appendix J), so the parts-like structure tracks the CP parameterization, not the step count. This is suggestive of the "parts-

based" representations sought in interpretability, though quantitative disentanglement metrics remain future work.

## 6.4 Do CP Models Learn the Same Surfaces?

The two extensions meet in a final question: are the low-rank surfaces that eigendecomposition *discovers* in dense models the same objects that CP *enforces* during training? QFS answers this directly from weights: we recompute both checkpoint families' per-class interaction matrices in input space under a single convention (CoM-free MNIST checkpoints, 5 seeds per family, $k = 20$) and compare all class pairs.[4] Class correspondence is clearly detectable: for dense weight-decay vs. Lambda CP ($R = 256$), seed-averaged same-class QFS is 0.18 against an off-diagonal floor of $-0.02$ (one-sided permutation test over class labels, $p = 1.0 \times 10^{-4}$, the smallest value $10^4$ resamples can produce; same-class minimum exceeds off-diagonal maximum in 22/25 seed pairs), and $k = 20$ captures $\geq 96\%$ of Frobenius energy in both families (Figure 6). Yet the correspondence sits far below every same-family *anchor*—our term for the reference QFS between models within one family: independently seeded dense models agree at 0.86, CP models with each other at 0.69, and even dense models at opposite regularization extremes (weight decay 0 vs. 1.0, bracketing CP's 0.1) agree at 0.78—so regularization strength alone cannot explain the gap. The gap is robust in form. It persists untruncated (full-rank same-class QFS 0.17) and under per-model centering—subtracting each model's class-mean interaction matrix, which removes any shared component that softmax ignores—ruling out truncation and gauge artifacts (0.18). It is, however, metric-dependent in size: reweighting pixel directions by their test-set variance (a data-whitened metric) raises same-class similarity to 0.72 (same-family anchors 0.96–0.98), and cross-family test-set logit correlation is 0.81 vs. 0.98 within-family; the analysis scripts and full grids are in the released results. The decisive test, however, is a *budget-matched control*: our CP models train full-batch ($\sim 29\times$ fewer optimizer steps than the dense mini-batch protocol), so we additionally trained dense models under the CP protocol exactly (full-batch, 100 steps; test accuracy 92.4%, below CP's 93.8%, with effective rank only 6.1—CP reaches dense-like effective rank at a budget where dense models have not yet developed their spectral structure). These budget-matched dense models diverge from the fully trained dense surfaces *at least as much* as CP does (same-class QFS 0.13 vs. CP's 0.18) while agreeing among themselves (0.84). A second control shows the gap is not specific to budget either: two *fully trained* dense families differing only in regularizer type (weight decay vs. noise augmentation) agree at only 0.16—the same magnitude as dense vs. CP (all pairwise grids in the released `qfs_cp_bridge.json`). We conclude that same-class QFS gaps of this size are generic across training regimes, and the divergence is attributable to training regime—budget and regularizer family—rather than to the CP constraint specifically: under matched budgets, CP models are slightly *more* accurate than dense and sit no farther from the fully trained surfaces than either control. Training CP under the dense mini-batch protocol exactly ($\sim 2{,}900$ optimizer steps; both weight-decay arms, 0.1 and 1.0; 5 seeds each; released as `qfs_cp_bridge_minibatch.json`) closes the loop. Fully trained CP models reach dense-level accuracy ($97.81 \pm 0.07\%$ vs. dense 97.49%) at dense-like effective rank (15.3), and their surfaces substantially coincide with the fully trained dense surfaces: same-class QFS 0.71 (0.75 in the wd=1.0 arm) against cross-seed anchors of 0.79 (mini-batch CP) and 0.86 (dense), with complete separation in 25/25 seed pairs (permutation $p = 10^{-4}$)—while sitting far from the full-batch CP family (0.18), confirming that family was budget-limited. At convergence, the enforced and the discovered structure substantially coincide: the divergence analyzed above was a training-budget effect, and enforcing interpretability costs neither accuracy nor fidelity to the dense solution. Methodologically, QFS resolves solution families cleanly from weights alone, and without matched controls a weight-space comparison misattributes training-regime effects to architecture—a cautionary result for such comparisons generally. Two questions stay open: the discovered and enforced objects are not the same algebraic species (symmetrized eigendecomposition vs. non-symmetric CP factors; the symmetric-CP variant of Section 7.5 would remove the mismatch), and whether the parts-like factor localization of Section 6.3 persists at full training is untested.

---

[4]CP tensors are reconstructed from the stored factors following the repository's `decompose` convention, which omits the column normalization applied in the forward pass; a functionally exact reconstruction (verified to reproduce CP logits to machine precision) changes every QFS value reported here by less than 0.001.

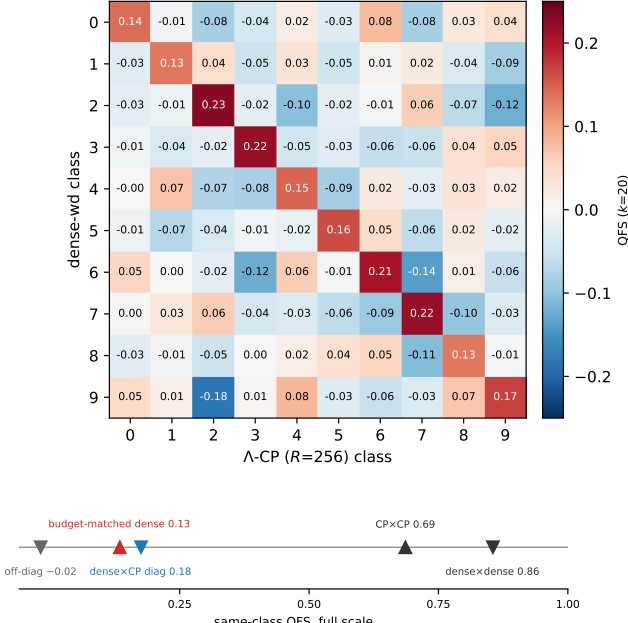

Figure 6: **QFS bridge between discovered and enforced structure.** Seed-averaged QFS ($k = 20$) between per-class decision surfaces of dense weight-decay and Lambda CP ($R = 256$) MNIST models (25 seed pairs). The diagonal (same class, mean 0.18) clearly exceeds the off-diagonal floor ($-0.02$; permutation $p = 1.0 \times 10^{-4}$) but sits far below the cross-seed anchors, shown on the full-scale anchor axis beneath the matrix (dense×dense 0.86, CP×CP 0.69). A budget-matched dense control diverges at least as much (same-class QFS 0.13), so the gap reflects training budget rather than the CP constraint (see text).

### 6.5 Conclusion

CP-decomposition yields qualitatively more localized, parts-like factors than dense eigendecomposition (quantitative disentanglement validation left for future work) and a compact analysis object: the inter-action tensor admits an $\mathcal{O}(R \times d_{\text{in}})$ description in CP form versus $\mathcal{O}(d_{\text{out}} \times d_{\text{in}}^2)$ dense—∼404k values at the featured $R = 256$ versus 6,146,560, and 39,475 at the matched-budget $R = 25$ of Appendix K. On efficiency: our CP and dense runs used different training protocols (full-batch, 100 optimizer steps, vs. mini-batch, ∼2,900 steps with a shuffling data loader), so their wall-clock times (∼1.8 s vs. ∼50 s per run, Table 16) measure the protocols, not the architectures. The correct form of the efficiency claim is that CP prices the bilinear interaction *linearly in the rank*: the interaction term costs $\mathcal{O}(R \cdot d_{\text{hidden}})$ parameters and multiply–accumulates versus the dense layer's $\mathcal{O}(d_{\text{hidden}}^2)$, so at $R = 32$ the interaction is ∼5× smaller (27,168 vs. 133,632 parameters in our implementation) at 92.7% accuracy (Table 14), and matched-protocol training steps are measurably faster at low rank (∼1.4× at $R \leq 32$), while at the featured $R = 256 = d_{\text{hidden}}$ there is no saving. (End-to-end savings in our implementation are capped by the shared input embedding, which dominates the parameter count.) These results are preliminary and limited to MNIST-scale experiments; Lambda and Gated variants provide a middle ground between strict rank constraints and dense regulariza-tion. Together, the two extensions give the transparency thesis an operational form: bilinear weight-space objects are *identifiable* (CP's uniqueness, Appendix K), *portable* across datasets (Section 5), and *enforceable* at training time—with enforcement recovering the discovered surfaces at convergence, at no accuracy cost (Section 6.4).

## 7 Discussion

### 7.1 Practical Challenges

Vision experiments were straightforward (∼1 min/run on our hardware, Table 16; std <1% across 40 runs). Language challenges included the SAE availability gaps of Section 2.3, ambiguous definitions (correlation

measure, "active" threshold), and memory constraints; the resulting threshold sensitivity is quantified in Table 3.

## 7.2 Environmental Impact

Total footprint: 0.169 kg CO2 across 224 runs (Table 16, Appendix L).

## 7.3 Lessons for Interpretability Research

This study yields four lessons that we believe apply well beyond Pearce et al. (2025).

**Lesson 1: Interpretability metrics are functions of auxiliary-model convergence, not just of the model being interpreted.** The most plausible driver of our language-reproduction gap is SAE training duration: the same model, features, and metric yield rank-2 correlations differing by a factor of 2.8 depending only on the SAE checkpoint used (Appendix C; measured on the layer-12 checkpoint series—a different layer and expansion factor than the headline analysis—so we treat it as strong evidence of sensitivity rather than a causal decomposition of our specific gap). Any SAE-based claim inherits this dependence, yet SAE training budgets are rarely reported (Bricken et al., 2023; Cunningham et al., 2023); we recommend that papers whose metrics depend on SAEs report SAE convergence curves for those metrics, and treat any single-checkpoint number as a lower bound.

**Lesson 2: Reproducibility of interpretability results is bottlenecked by artifact availability, not code.** The original authors released code, models, and many SAEs—far more than is typical—and their vision results reproduce fully. The language gap arose because *specific* artifacts ($4\times$ expansion SAEs at particular layers, SAE training configurations) were missing; interpretability pipelines are too sensitive for "nearest available substitute" to be a faithful proxy (Section 2.3). Releasing the base model is not enough: the auxiliary artifacts *are* part of the experimental configuration.

**Lesson 3: Claims of the form "most features are X" are not falsifiable as stated.** The paper's central language claim depends on an unspecified quantifier ("most"), correlation variant, and activity threshold for feature inclusion; our sensitivity analysis (Table 3) shows the verdict flips between "reproduced" and "not reproduced" within reasonable readings of these choices. Quantitative interpretability claims should specify the statistic, threshold, and population precisely enough that an independent team can compute the same number. Our own results illustrate the danger from the inside: fw-medium's verdict against the original 69% flips from "fails" (44.5%) to "matches" (73.4%) depending solely on whether rarely-active dictionary features—which only fw-medium's larger dictionary contains—are included in the population (Section 4.2).

**Lesson 4: Weight-based analysis proved more reproducible than activation-based analysis— which supports the original paper's own thesis.** Every claim computable purely from weights (eigen-decomposition, effective rank, low-rank truncation) reproduced cleanly, whereas the *quantitative prevalence* claims routed through data and auxiliary models (SAE rank-2 correlations) reproduced only partially—the qualitative SAE-based circuit findings did survive (Table 1). The two explanations are confounded: weight-based analyses were also the ones we could retrain from scratch, while SAE-based analyses depended on third-party artifacts (Lesson 2), so we cannot fully separate "weights are more robust" from "artifacts were missing." Still, the direction of the evidence is consistent with the paper's core argument: interpretability grounded in weights removes the very degrees of freedom (evaluation data, auxiliary-model training) that made the language results fragile. Weight decay buys this transparency cheaply—effective rank drops from 38.5 to 15.5 at <1% accuracy cost—a trade-off we would recommend by default for models of this scale.

## 7.4 Communication with Original Authors

We did not communicate with the original authors; all experiments used publicly available code and pre-trained artifacts.

### 7.5 Limitations

- **SAE artifact mismatch.** The $4\times$ expansion SAE the paper's configuration requires is unavailable at the relevant layer for `fw-medium` (Section 2.3), preventing exact reproduction and likely affecting correlation results.
- **SAE training duration.** Correlation quality depends on how long the SAE was trained, which the paper does not specify; the public checkpoints may not match it.
- **Qualitative interpretability assessment.** Eigenvector quality ("digit-like patterns") is evaluated visually; we provide no quantitative measure of visual interpretability.
- **CP-decomposition scope.** CP results are limited to MNIST with a single bilinear hidden layer; generalization to other datasets, architectures, or scales is not tested. Extending CP-as-architecture to language is structurally harder than an added experiment: because the constraint applies *during training*, it requires pretraining bilinear transformers with CP-constrained MLP blocks from scratch, then training SAE suites on the resulting models—an undertaking we scope as future work. (A minimal no-SAE variant—training a small CP-constrained transformer and reporting perplexity—would test trainability but none of the interpretability claims this report is about.) Alternative factorizations also remain unexplored: symmetric CP (matching the symmetrized analysis object exactly), block-term decompositions interpolating between CP and Tucker, and tensor-train formats for multi-layer stacks.

### 7.6 Reproducibility Statement

Code, configuration files, and result artifacts are available at `https://anonymous.4open.science/r/bilinear_mlp_reproduction-CE76`; dataset and HuggingFace artifact details are in Appendix A, hardware and runtime in Appendix L.

## 8 Conclusion

The weight-based interpretability claims of Pearce et al. (2025) hold up well under independent reproduction. Vision results reproduce fully: weight decay induces low-rank, interpretable eigenstructure at negligible accuracy cost—our ablations identify it, not noise augmentation, as the dominant cause. Language results are a constrained replication: AND-gate negation circuits are confirmed, but the prevalence of low-rank feature interactions holds for only one of three public models (32%–65% of features at the original threshold), a gap we trace to SAE convergence, model training compute, and missing public artifacts. Our extensions strengthen the transparency thesis from two directions: learned eigenstructure transfers across datasets as genuinely structural features (quantified by Quadratic Form Similarity where cosine similarity fails), and the low-rank structure discovered post-hoc can be enforced during training via CP-decomposition—with fully trained CP models converging to the dense surfaces at dense-level accuracy (Section 6.4): enforced and discovered structure coincide at convergence. Together, the extensions say not only *that* weight-based interpretability works here but in what sense and on which measures: the low-rank, interpretable structure is a property of the converged solution itself—portable across datasets (as measured by QFS on the decision surfaces) and reachable equally by post-hoc discovery or architectural enforcement—rather than an artifact of the analysis method or of any particular parameterization. The broader takeaway (Section 7.3): what reproduces cleanly is precisely what depends only on weights, suggesting the field's reproducibility problems are concentrated in its auxiliary-model pipelines and that weight-based methods deserve attention for that reason alone. Future work should develop quantitative interpretability metrics for eigenvector quality and test CP-decomposition beyond MNIST scale.

### Broader Impact Statement

This work contributes to mechanistic interpretability, aiming to make neural networks more transparent; improved interpretability methods could help identify biases, failure modes, and unexpected behaviors in deployed AI systems. Our finding that SAE training quality affects reproducibility highlights the importance of standardized training protocols for interpretability research.

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

## A  Hyperparameter Details

**Vision Experiments.** Architecture: 1-hidden-layer bilinear MLP, $d_{\text{hidden}} = 256$. Training: 100 epochs, batch size 2048, AdamW optimizer, lr $10^{-3}$ with cosine annealing. Seeds: 42–46 (5 seeds per configuration).

**Language Experiments.** Models: `ts-medium` (6L, 512d, 29M params), `fw-small` (12L, 768d, 162M params), `fw-medium` (16L, 1024d, 334M params). SAEs: TopK sparsity ($k = 30$), expansion ratios 4–8× (the SAE-training-duration analysis in Appendix C additionally uses 16× checkpoints). Correlation analysis: features with a valid Pearson correlation per model (n=2,048 / 3,071 / 5,278); fw-medium has 5,688 total active features, of which 410 yield no valid correlation (rarely-firing or constant-activation features) and are excluded.

## B  Vision Results

This appendix provides visualizations from our reproduction of the paper's vision experiments (Section 4).

### B.1  Eigenspectrum Analysis

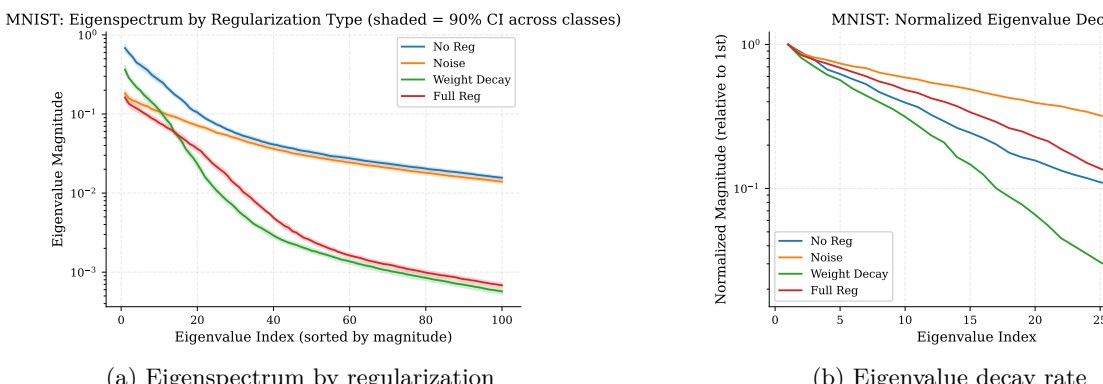

(a) Eigenspectrum by regularization

(b) Eigenvalue decay rate

Figure 7: Eigenspectrum analysis on MNIST. Weight decay produces the steepest spectral decay, concentrating variance in 10–20 eigenvectors.

## B.2  Cross-Seed Consistency and Truncation (Paper Figure 5)

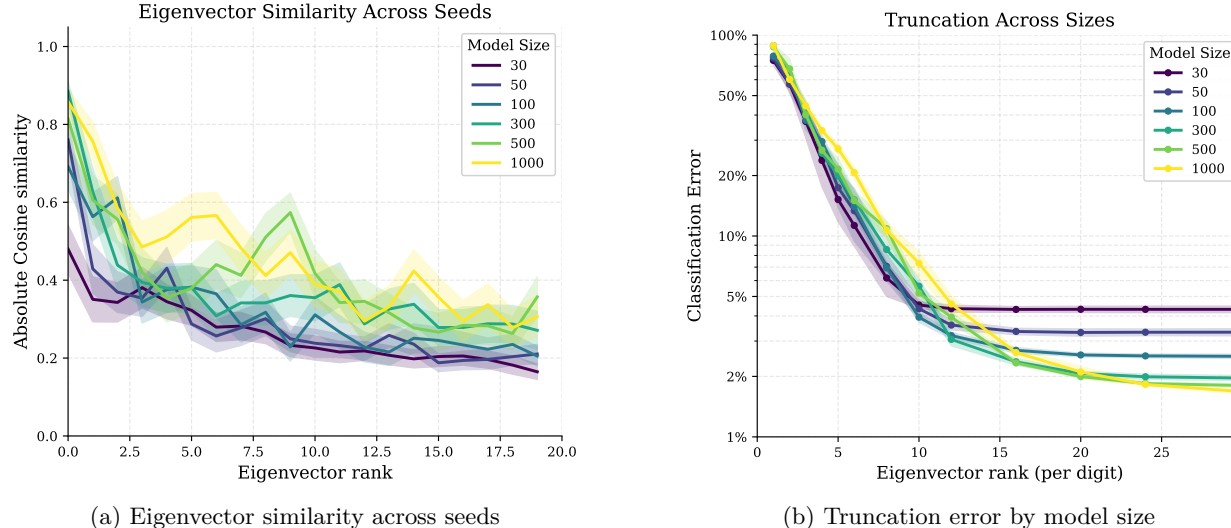

(a) Eigenvector similarity across seeds

(b) Truncation error by model size

Figure 8: Paper Figure 5 reproduction. **(a)** Top eigenvectors are highly consistent across random seeds, especially for larger models, with cosine similarity decreasing smoothly as eigenvector rank increases. **(b)** Truncation error decreases with model size; keeping ∼20 eigenvectors per class is sufficient for near-full accuracy.

## B.3  Noise Effect (Paper Figure 4)

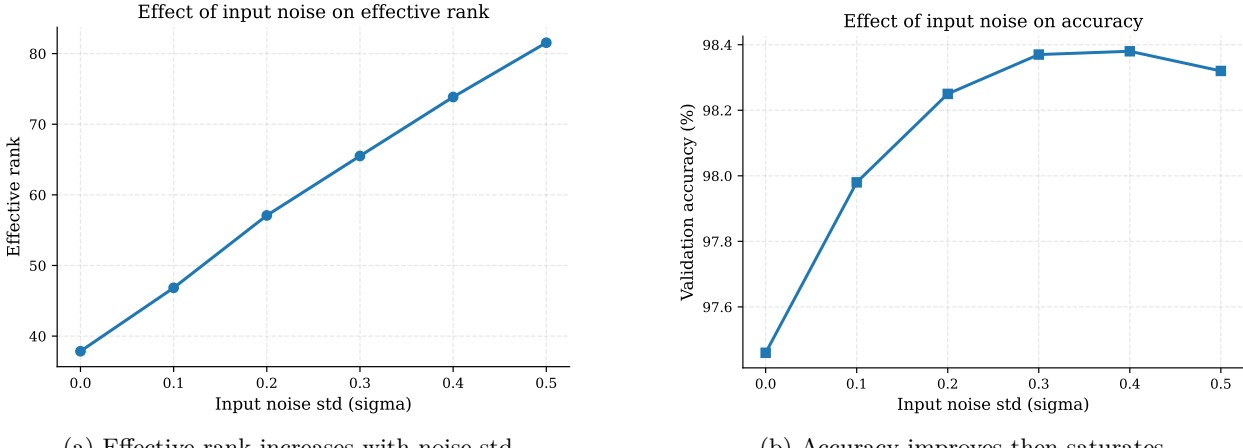

(a) Effective rank increases with noise std

(b) Accuracy improves then saturates

Figure 9: Noise augmentation effect on effective rank and accuracy (reproducing paper Figure 4). Higher noise spreads variance across more eigenvectors while improving generalization. Notation note for this appendix: embedded figure titles use $\lambda$ for weight decay (legacy plot titles); the text reserves $\lambda$ for eigenvalues.

Figure 10 shows how the top eigenvector for digit 5 changes across noise augmentation levels. Higher noise produces more diffuse, averaged eigenvectors that better capture the canonical digit shape.

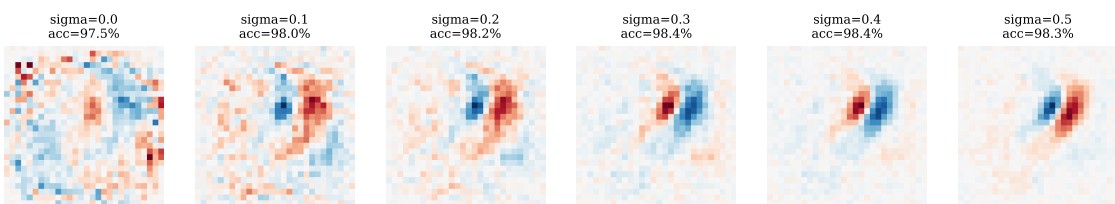

Figure 10: Top eigenvector for digit 5 across noise levels ($\sigma = 0.0$ to $0.5$). Accuracy improves with noise despite spreading variance across more eigenvectors.

## B.4 Challenge Task (Paper Figure 6)

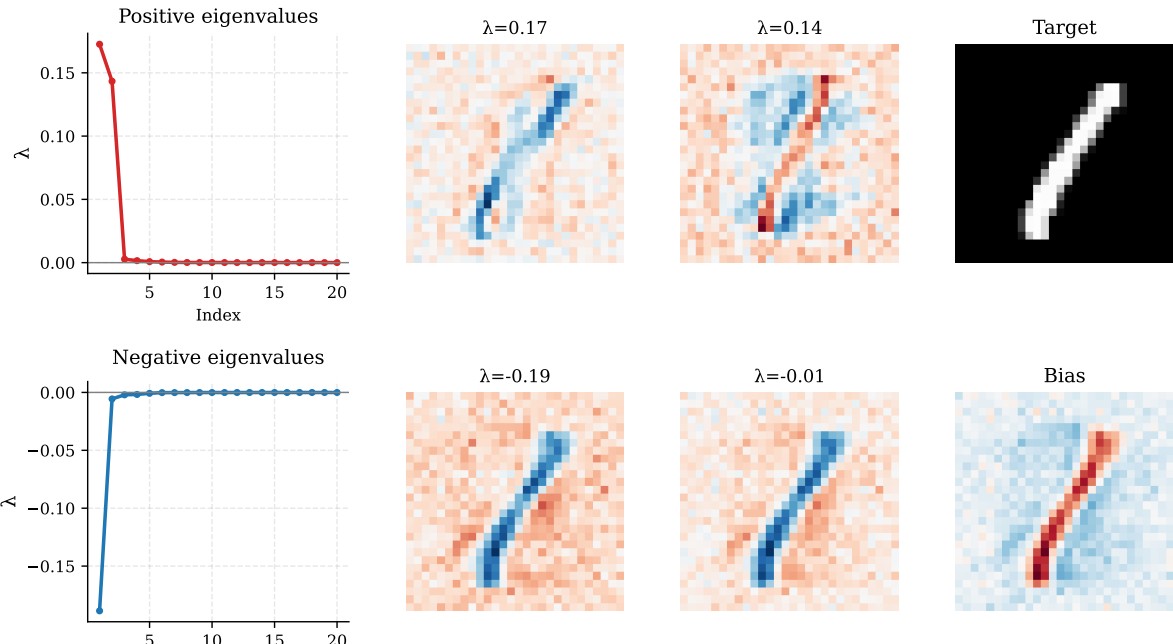

Figure 11: Challenge task eigendecomposition with full regularization (noise $\sigma = 0.5$, weight decay $= 1.0$), reproducing paper Figure 6. **Left:** Eigenvalue decay shows dominant positive/negative modes. **Middle:** Top eigenvectors resemble the target digit pattern. **Right:** Target digit and bias visualization.

We trained challenge task models under four regularization configurations to analyze how regularization affects the eigendecomposition structure. Figure 12 compares eigenvalue decay across these variants.

Challenge task: eigenvalue decay by regularization

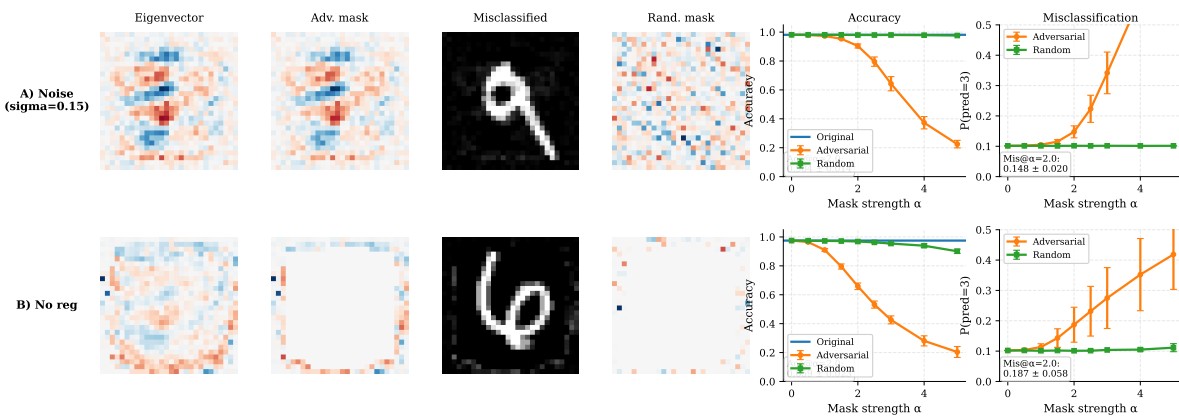

Figure 12: Challenge task eigenvalue decay by regularization configuration. Weight decay produces the sharpest spectral decay, concentrating information in fewer eigenvectors.

## B.5 Adversarial Perturbations (Paper Figure 7)

Figure 13: Adversarial mask analysis (reproducing paper Figure 7). **Row A:** Noise-regularized model ($\sigma = 0.15$). **Row B:** No regularization (with rare-edge constraint). Eigenvector-derived masks cause significantly larger accuracy drops than random masks of equal magnitude.

## B.6 Eigenvector Quality Comparison

Figure 14 shows the effect of noise-only regularization on eigenvector interpretability.

MNIST (Noise only: sigma=0.5, λ=0.0): Top Eigenvectors

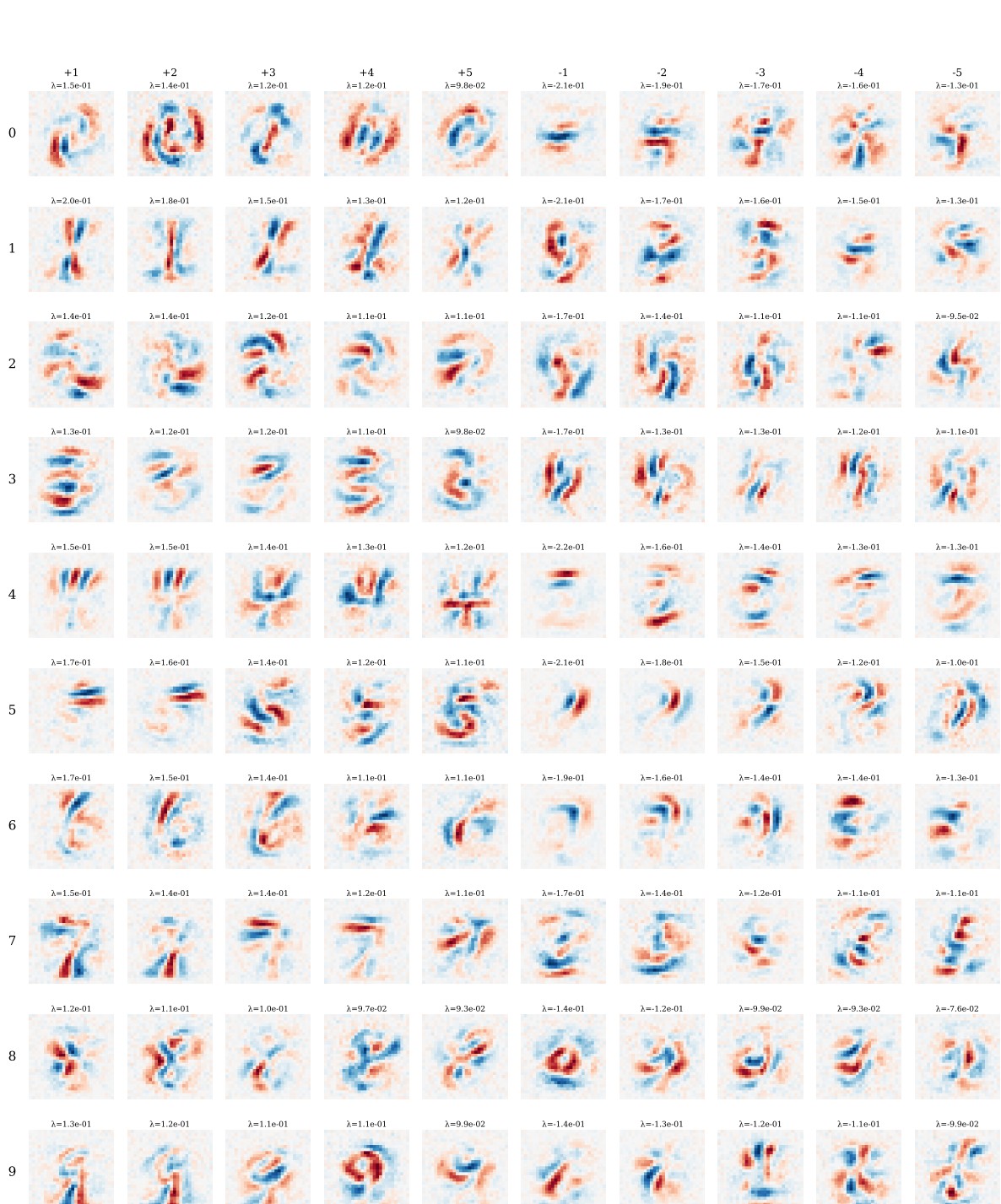

Figure 14: Top eigenvectors with noise augmentation only ($\sigma = 0.5$, $\lambda = 0.0$). Noise alone produces recognizable digit patterns, though less sharp than with weight decay.

## B.7 Cross-Dataset Comparison: MNIST vs Fashion-MNIST

Figure 15 shows eigenvector quality on Fashion-MNIST under noise-only regularization.

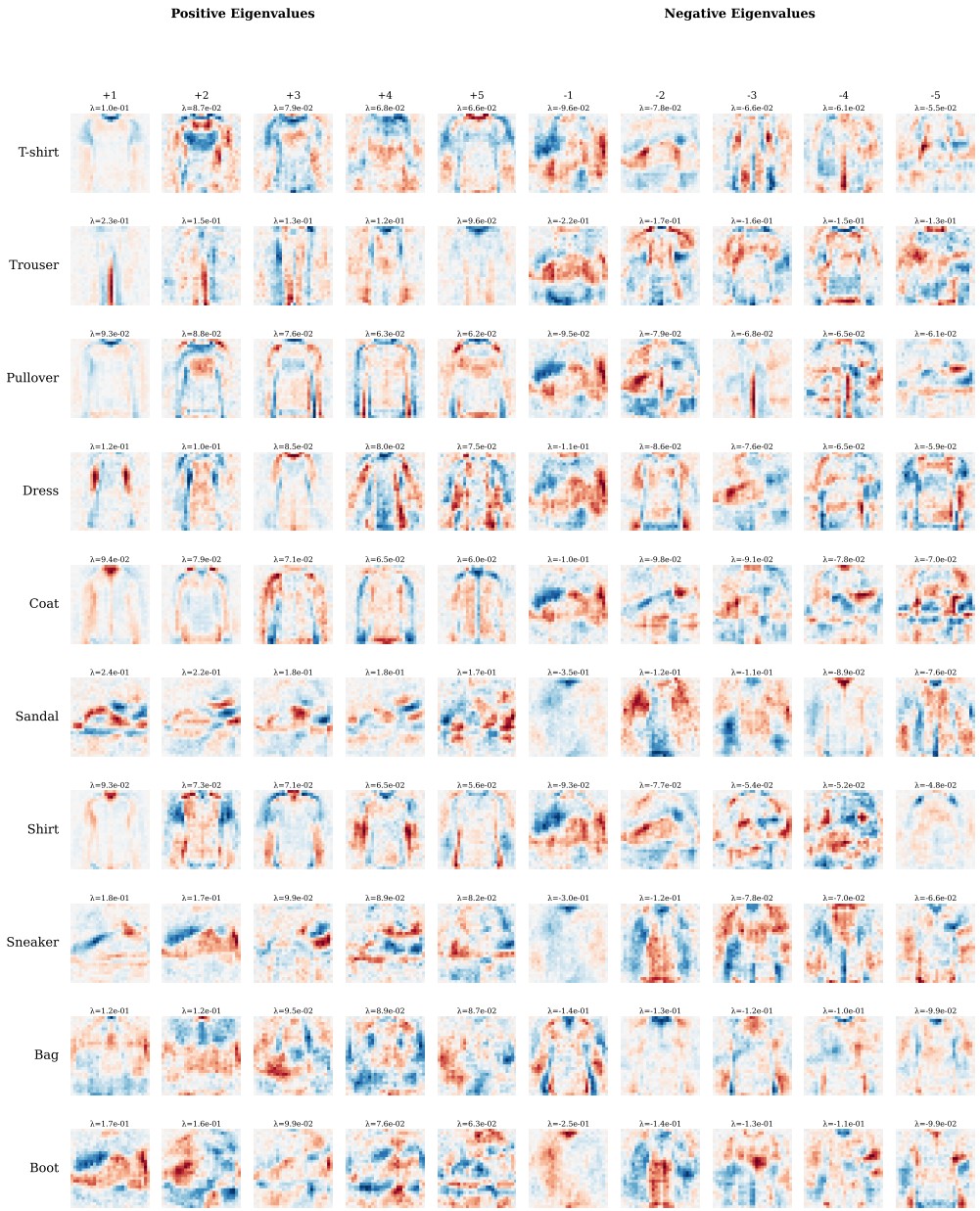

Figure 15: Fashion-MNIST top eigenvectors with noise augmentation ($\sigma = 0.5$). Class labels: T-shirt, Trouser, Pullover, Dress, Coat, Sandal, Shirt, Sneaker, Bag, Boot.

The silhouette-like patterns for clothing items (T-shirt, Dress, Coat) demonstrate that the eigenvector interpretability extends beyond digit recognition to more complex shape categories.

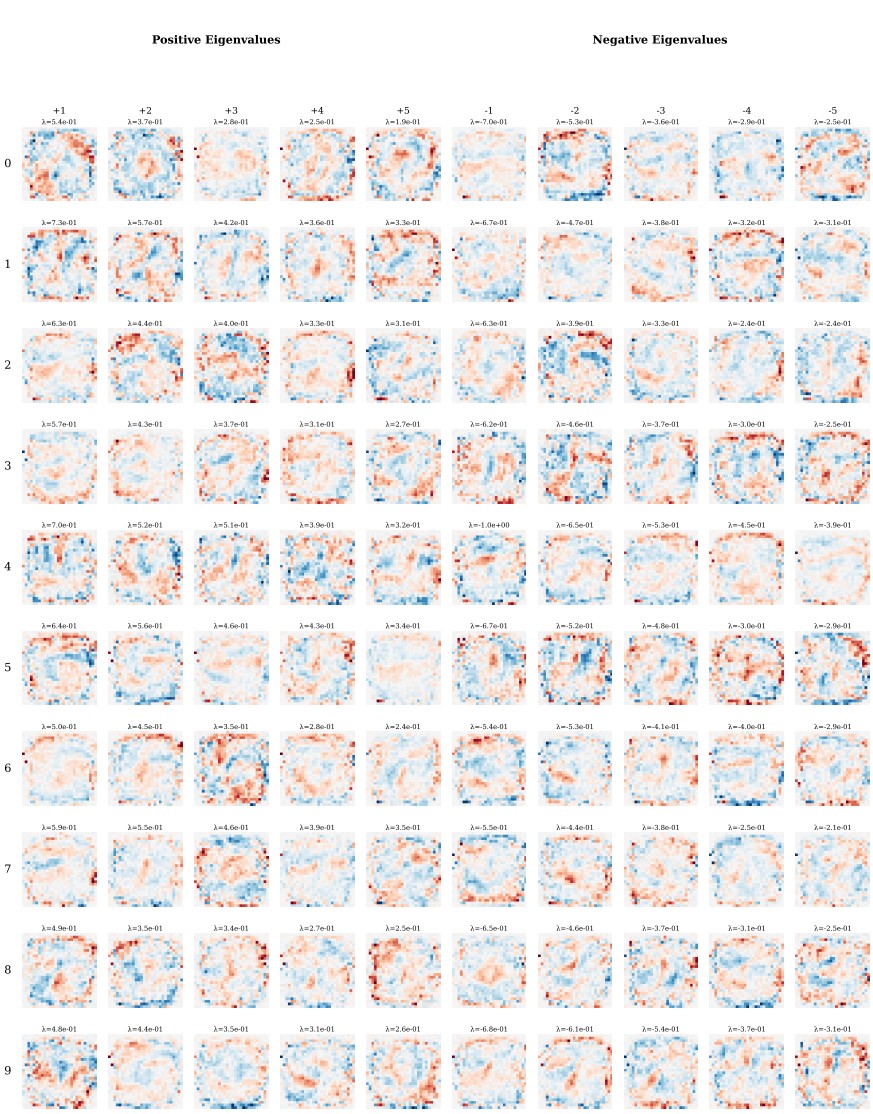

Figure 16: Top eigenvectors for each digit class (0–9) without regularization: diffuse, noise-like patterns (V3).

Figure 17: Top eigenvectors for each digit class (0–9) with regularization (noise $\sigma = 0.5$, weight decay $= 1.0$): recognizable digit strokes (V2).

## C  Language Experiment Results

Table 4: Language model configurations for correlation analysis (Figure 9).

| Model | Layers | Params | SAE Layer | Expansion | $k$ | Dataset |
|---|---|---|---|---|---|---|
| ts-medium | 6 | 29M | 4 | 4 | 30 | TinyStories |
| fw-small | 12 | 162M | 8 | 4 | 30 | FineWeb-EDU (FineWeb subset) |
| fw-medium | 16 | 334M | 10 | 8 | 30 | FineWeb-EDU (FineWeb subset) |

The original paper's Figure 8 analysis (our Section 4.2) uses fw-medium at layer 7 with expansion 8 (public SAEs available at that layer); our reproductions of the paper's Figures 8 and 9 appear in the main text as Figures 2 and 3.

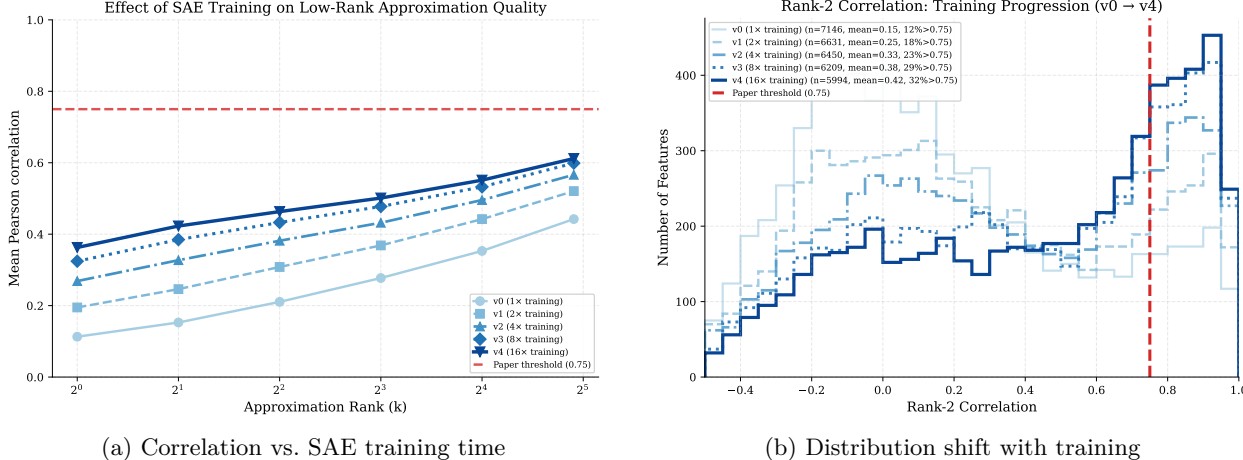

(a) Correlation vs. SAE training time        (b) Distribution shift with training

Figure 18: Effect of SAE training duration on interpretability (fw-medium layer 12, expansion 16, active features only). Correlation distributions shift rightward with increased SAE training, consistent with the paper's qualitative claim that better-trained SAEs yield stronger low-rank predictability. This analysis is computed on a fixed FineWeb-EDU sample to enable a controlled comparison across SAE checkpoints.

# D   Sentiment Negation Circuit Investigation

This appendix presents our detailed investigation of AND-gate circuits in bilinear transformers.

## D.1   SAE Availability Investigation

We investigated the publicly available SAEs on HuggingFace to determine reproducibility constraints.

**ts-medium-scope** (paper's "ts-tiny", 6 layers):

- Layers 0–4: Only `mlp-out`, `resid-mid`, and `resid-pre` SAEs (hook points: MLP output, residual stream after attention, and residual stream before the block, respectively)
- Layer 5: `mlp-in` and `mlp-out` available
- **Critical: No `mlp-in` SAE at layer 4** (the layer used in the paper's Figure 8)

**fw-medium-scope** (16 layers):

- All layers 0–15: `mlp-in` and `mlp-out` at 8× expansion
- 4× expansion exists only at layers 6 and 12 (`mlp-out`) and layer 1 (`resid-mid`)—in particular, *not* at layer 10, the 2/3-depth layer the correlation analysis requires
- Layer 12: Additional v0–v4 training-time variants at 16× expansion

Since the circuit analysis requires *both* `mlp-in` and `mlp-out` SAEs (the original repository's `Tracer` analysis class expresses the interaction between input-side and output-side SAE features), the paper's Figure 8 configuration (ts-tiny, layer 4) cannot be reproduced with public artifacts. We therefore use fw-medium (layer 7, expansion 8) which has both SAE types available. The original authors' language tutorial (`tutorials/2_language.ipynb` in their code release; "the tutorial" throughout this appendix) also uses fw-medium with features 3834/751, confirming this is a valid alternative.

## D.2   AND-gate Circuit Analysis

The following analysis uses fw-medium (16 layers, 334M parameters, trained on FineWeb-EDU) with layer 7 SAEs at 8× expansion (8,192 output features).

### D.2.1 AND-gate Circuit Definition

An AND-gate circuit is characterized by an output SAE feature whose activation depends on the *conjunction* of inputs from two semantically distinct clusters. Mathematically, for output feature $c$, the activation is:

$$z_c(x) = x^\top Q_c x \approx \lambda_1 (v_1^\top x)^2 + \lambda_2 (v_2^\top x)^2 \tag{8}$$

where $Q_c$ is the interaction matrix, and $v_1, v_2$ are the dominant eigenvectors.

Strong AND-gate behavior requires:

1. **Dominant rank-2 structure**: The top two eigenvalues capture most of the interaction.
2. **Opposing eigenvalue signs**: $\lambda_1$ and $\lambda_2$ have opposite signs, creating cancellation when only one cluster is present.
3. **Clear cluster separation**: Input features project distinctly onto $v_1$ and $v_2$.

### D.2.2 AND-score Metric

We quantify conjunction structure by the ratio of cross-cluster to within-cluster interaction strength. Partitioning a feature's top input features into its two clusters $C_1, C_2$ and restricting $Q_c$ accordingly,

$$\text{AND-score} = \frac{\text{mean}_{i \in C_1, \, j \in C_2} |Q_{ij}|}{\frac{1}{2} \left( \text{mean}_{i,j \in C_1} |Q_{ij}| + \text{mean}_{i,j \in C_2} |Q_{ij}| \right)}. \tag{9}$$

High values indicate that the feature's activation is driven by the *joint* presence of both clusters (cross terms) rather than by either cluster alone (within terms), which is the defining property of an AND gate.

### D.2.3 Search Procedure

We analyzed all 8,192 output features in fw-medium layer 7:

1. Computed the interaction matrix $Q_c$ for each feature $c$ and its eigendecomposition $Q_c = \sum_j \lambda_j v_j v_j^\top$
2. Shortlisted features by top eigenvalue magnitude (the tutorial's heuristic, as in Section 4.2)
3. Clustered each shortlisted feature's top input features and computed the cross-to-within AND-score above
4. Analyzed the top-ranked circuits for semantic interpretation

## D.3 Results: Feature 3834 vs Feature 751

### D.3.1 Tutorial Feature 3834

Feature 3834 (used in the paper's tutorial) encodes "not-bad" contexts with weaker AND-gate structure: diffuse eigenvalue spectrum and less distinct cluster separation.

### D.3.2 Best Circuit: Feature 751

Our search identified feature 751 as having the strongest AND-gate structure (Table 6).

## D.4 Semantic Interpretation of Feature 751

The input clusters have clear semantic content based on token analysis:

**Positive cluster** (3 features: 5212, 7655, 253):

- Feature 5212: "grateful", "opportunity", "Luckily", "thanks"
- Feature 7655: "beauty", "lovely", "amazing", "enjoy", "fun"
- Feature 253: "couldn't help" (compulsion/emotion contexts)
- Semantics: Positive sentiment/gratitude/appreciation

**Negative cluster** (5 features: 5201, 3601, 6051, 6921, 1003):

- Feature 5201: "unfortunate", "bad"
- Feature 3601: "problems", "disease"
- Feature 6051: "overly", "too narrow"
- Feature 6921: "no need"
- Feature 1003: "reducing", "stopping", "stop"

**Circuit function:** Feature 751 detects contexts where negative concepts appear *without* positive modulation. The AND-gate structure arises because:

1. The top eigenvalue (negative) suppresses activation when positive cluster projects onto $v_1$
2. The second eigenvalue (positive) activates when negative cluster projects onto $v_2$
3. When both clusters are present, the opposing signs cause partial cancellation

This confirms the mechanism: high activation requires negative concepts *without* positive sentiment, consistent with a "problematic situation" detector.

## D.5  Linear Subspace Structure

Following the tutorial methodology, we compute cosine similarity between SAE decoder directions for the top output features with high eigenvalues. Features 3834 and 751 have decoder cosine similarity $-0.73$—strongly anti-aligned—which the tutorial calls a "somewhat linear subspace." The features are semantically contrasting (not-bad vs. not-good); the analogous pair the paper reports in ts-medium is more extreme (decoder cosine $-0.975$), and the two features' top eigenvectors are nearly anti-aligned (cosines $-0.996$ and $-0.990$). Semantic interpretation comes from inspecting top-activating tokens: feature 3834 fires on negated negative words, while feature 751 fires on negated positive words. The paper's tutorial observes: "It's not surprising that this forms a linear subspace as the two are opposites."

## D.6  Cross-Dataset Validation

As additional validation beyond the paper, we test the same circuit on FineWeb-16k (the model's training distribution) in addition to TinyStories. Figure 19 shows the results, and Table 5 provides a quantitative comparison.

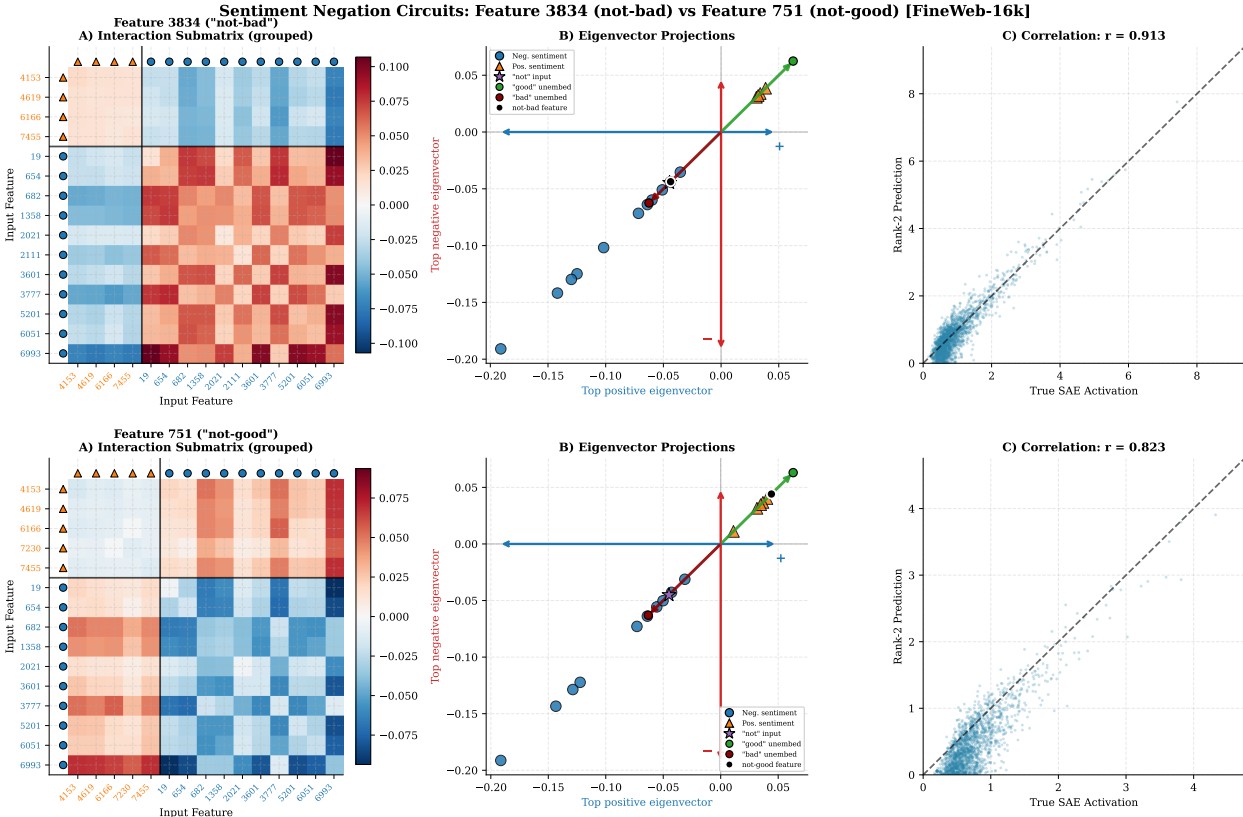

Figure 19: AND-gate circuit on FineWeb-16k (model's training distribution). Correlations: 0.91 (feature 3834), 0.82 (feature 751). Feature 751 activates more frequently on FineWeb-16k (6,517 vs 2,542 samples), likely due to more "not-good" contexts in educational/scientific text.

Table 5: Cross-dataset comparison of negation circuit correlations.

| Feature | TinyStories | FineWeb-16k | $\Delta$ |
|---|---|---|---|
| 3834 (not-bad) | 0.931 (n=9,719) | 0.913 (n=9,937) | $-0.018$ |
| 751 (not-good) | 0.831 (n=2,542) | 0.823 (n=6,517) | $-0.008$ |

The correlations remain strong across both datasets, with differences of only 1–2%. This demonstrates that the discovered circuit is robust to distribution shift and is a genuine property of the bilinear architecture, not a dataset artifact.

## D.7 Summary

Table 6: Comparison of AND-gate circuit properties.

| Property | Feature 3834 | Feature 751 |
|---|---|---|
| Rank-2 correlation (TinyStories) | 0.931 | 0.831 |
| Rank-2 correlation (FineWeb-16k) | 0.913 | 0.823 |
| Decoder cosine similarity | \multicolumn{2}{c}{$-0.73$ (strongly anti-aligned)} | |
| Semantic interpretation | "not-bad" | "not-good" |

**Key findings:** (1) Features 3834 and 751 are semantically contrasting negation features (not-bad vs. not-good) with near anti-parallel decoder directions (cosine $-0.73$). (2) The AND-gate circuit (negation $\otimes$

sentiment → negated-sentiment) is confirmed for both features. (3) Cross-dataset validation shows the circuit is robust to distribution shift (TinyStories vs. FineWeb-16k).

# E  Cross-Dataset Robustness Results

Figure 20: Three-way eigenvector comparison: MNIST-0, EMNIST-O, and EMNIST-X. The top eigenvectors for each class are visually distinct from one another.

Table 7: Cross-Dataset Accuracy: MNIST ↔ EMNIST-Digits (mean over 5 seeds).

| Direction | Accuracy |
|---|---|
| MNIST → EMNIST-Digits | $97.527 \pm 0.024\%$ |
| EMNIST-Digits → MNIST | $97.910 \pm 0.026\%$ |
| Bidirectional Average | $97.719 \pm 0.012\%$ |

Table 8: Cross-Dataset Accuracy: MNIST → USPS (mean over 5 seeds).

| Model | Accuracy |
|---|---|
| Baseline (no reg) | $53.42 \pm 0.88\%$ |
| Regularized | $72.33 \pm 0.83\%$ |

Table 9: Letter→Digit Transfer (regularized CoM model, 5 seeds).

| Letter → Digit | Accuracy |
|---|---|
| O → 0 | $99.60 \pm 0.05\%$ |
| I → 1 | $87.08 \pm 0.10\%$ |
| Z → 2 | $88.75 \pm 0.26\%$ |
| S → 5 | $87.38 \pm 0.22\%$ |

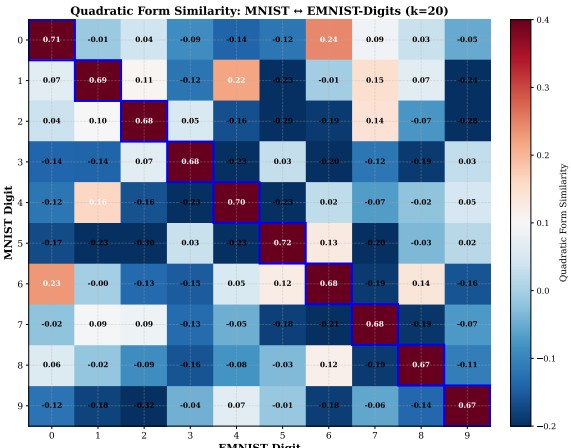

Figure 21: Quadratic Form Similarity ($k = 20$), MNIST vs. EMNIST-Digits. The strong diagonal confirms writer-independent circuits; the companion MNIST vs. EMNIST-Letters comparison is Figure 4b in the main text.

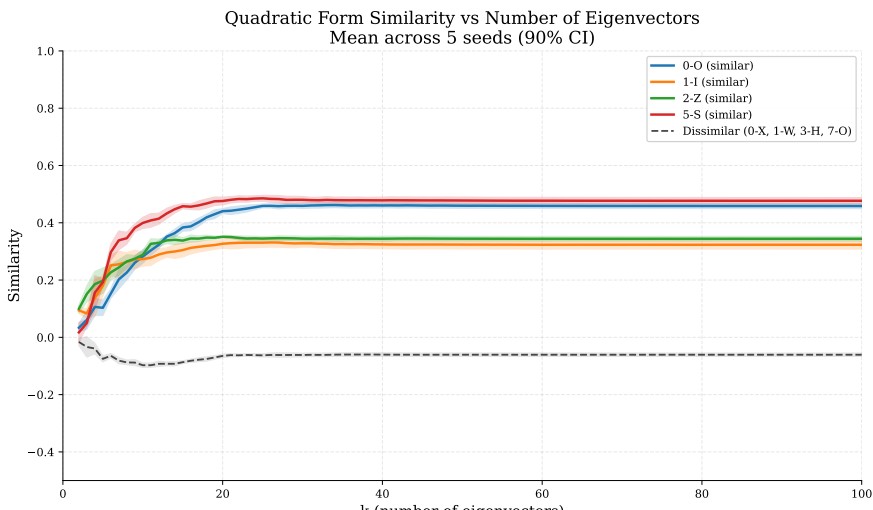

Figure 22: Quadratic Form Similarity vs number of eigenvectors $k$. Similar pairs converge around $k = 20$; dissimilar pairs stay near zero with non-overlapping 90% CIs.

Table 10: Ranking analysis: Position of expected letter among 26 EMNIST letters sorted by Quadratic Form Similarity. Random baseline: rank 13.

|  | 0–O | 1–I | 2–Z | 5–S | Mean |
|---|---|---|---|---|---|
| Rank | **1** | **2** | **1** | **1** | **1.2** |

## F  Cosine Similarity Limitations

Figure 4a (main text) shows the absolute cosine similarity heatmap between MNIST digit classes and EM-NIST letter classes: cosine similarity fails to produce a clear distinction between geometrically similar and dissimilar class pairs. Figure 20 provides a three-way eigenvector comparison that further illustrates this limitation.

Table 11 provides a detailed per-eigenvector breakdown (balanced top-10 eigenvectors per class, best-match absolute cosine, mean over 5 seeds): the similar pair 0–O scores $0.312 \pm 0.016$ while the dissimilar pair 0–X

scores $0.364 \pm 0.029$—cosine similarity not only fails to separate the pairs but slightly inverts the expected ordering, demonstrating its fundamental limitation for comparing bilinear eigensystems.

Table 11: Absolute Cosine Similarity between Top 10 Eigenvectors (Balanced: 5 Positive + 5 Negative), Mean over 5 seeds

| Eigenvector Rank | Eigenvalue Sign | Maximum Absolute Cosine Similarity | |
|:---:|:---:|:---:|:---:|
| | | MNIST-0 vs EMNIST-O | MNIST-0 vs EMNIST-X |
| 1 | + | $0.426 \pm 0.083$ | $0.337 \pm 0.053$ |
| 2 | + | $0.501 \pm 0.105$ | $0.713 \pm 0.072$ |
| 3 | + | $0.459 \pm 0.069$ | $0.539 \pm 0.058$ |
| 4 | + | $0.297 \pm 0.049$ | $0.267 \pm 0.150$ |
| 5 | + | $0.283 \pm 0.033$ | $0.256 \pm 0.073$ |
| 6 | - | $0.110 \pm 0.072$ | $0.364 \pm 0.112$ |
| 7 | - | $0.161 \pm 0.036$ | $0.368 \pm 0.159$ |
| 8 | - | $0.339 \pm 0.092$ | $0.319 \pm 0.105$ |
| 9 | - | $0.286 \pm 0.051$ | $0.225 \pm 0.080$ |
| 10 | - | $0.255 \pm 0.110$ | $0.248 \pm 0.064$ |
| **Mean** | | $\mathbf{0.312 \pm 0.016}$ | $\mathbf{0.364 \pm 0.029}$ |

## G    Quadratic Form Similarity Derivation

For a bilinear MLP, the class-$c$ logit is a quadratic form $y_c(x) = x^\top A_c x$ where $A_c = V_c \operatorname{diag}(\boldsymbol{\lambda}^{(c)}) V_c^\top$ is the symmetrized interaction matrix with its own eigenbasis $V_c$ per class.

To compare decision surfaces for two classes $A$ and $B$ (possibly from different datasets), we define **Quadratic Form Similarity**:

$$\operatorname{sim}(A, B) = \frac{\operatorname{tr}(A \cdot B)}{\|A\|_F \|B\|_F} \in [-1, 1]$$

For a rank-$k$ truncation using top eigenmodes, this expands to:

$$\operatorname{sim}(A, B) = \frac{\sum_{i,j} \lambda_i^A \lambda_j^B \cos^2(v_i^A, v_j^B)}{\|\lambda^A\|_2 \|\lambda^B\|_2}$$

Equivalently, since $\nabla^2 y_c = 2A_c$, QFS at the full-matrix level is the Frobenius cosine between the Hessians of the two class logits. QFS is invariant to orthogonal input transformations applied to both models, $\operatorname{sim}(RAR^\top, RBR^\top) = \operatorname{sim}(A, B)$, and to *positive* rescaling of either matrix (negative rescaling flips the sign), but deliberately *not* invariant to independent rotations of each model's basis, which change the computed function. The rank-$k$ expansion above is exact when the eigenvectors are orthonormal (as in the input-space eigendecompositions of Section 6.4); for the cross-dataset pipeline's eigenvectors—computed in the model's embedding space and projected to pixel space through the input embedding $w_e$, after which they are no longer orthonormal—it is an approximation, agreeing with the direct rank-20 matrix computation to at most 0.023 on the headline pair of Section 5.2.3.

**Key properties solving the cosine similarity problem:**

- Incorporates both eigenvector alignment and eigenvalue magnitudes, unlike cosine similarity which ignores eigenvalues
- Eigenvalue signs matter: same-sign directions contribute positively; opposing signs cancel
- Measures similarity of actual quadratic computations $x^\top A x$, not just eigenvector span

This enables discrimination between geometrically similar pairs (0–O: 0.437 at seed 42; mean over the four geometric pairs across seeds 0.399) and dissimilar pairs (0–X: $-0.093$ at seed 42; mean over control pairs $-0.057$), where eigenvector cosine similarity failed ($0.312 \pm 0.016$ for 0–O vs. $0.364 \pm 0.029$ for 0–X, Table 11).

## H  QFS vs. CKA Comparison

We compare QFS against linear CKA (Kornblith et al., 2019) on the class-pair discrimination task of Section 5.2.3. For class $c$, the rank-$k$ variant forms the spectral representation $R_c \in \mathbb{R}^{n \times k}$ with $R_c[i,j] = \sqrt{|\lambda_j|}\,(\mathbf{v}_j^\top \mathbf{x}_i)$ over a shared evaluation set ($k = 20$, matching the QFS truncation); the scalar variant applies CKA to the quadratic outputs $y_c(\mathbf{x}) = \mathbf{x}^\top A_c^{(k)} \mathbf{x}$, which preserves eigenvalue signs. The evaluation set is the union of the MNIST test set (10,000 images) and the EMNIST-Letters test set (20,800 images), CoM-normalized ($n = 30{,}800$).

Table 12 reports all four metrics for the four geometrically similar pairs and eight mismatched control pairs (seed 42; cross-seed statistics in the released results). The Cosine column is the mean cosine of the principal angles between the two classes' rank-$k$ eigenspaces ($k = 20$); it is a different statistic from the balanced top-10 best-match cosine of Table 11, which explains the different scales ($\sim 0.70$ vs. $\sim 0.31$ for the same pairs)—neither variant separates the geometric from the control pairs. Both metrics are tested identically (two-sample $t$-test of the 4 similar vs. 8 control pairs, per seed); because the 12 pairs share classes and are not independent, we regard complete separation (minimum similar > maximum control) as the more meaningful criterion than the $p$-values. QFS achieves complete separation in all five seeds, with $p < 10^{-4}$ in every seed. Scalar CKA ranks all four expected pairs first (QFS ranks 3/4 first) but its distributions overlap in every seed (0–X $\approx 0.50$ exceeds 2–Z $\approx 0.29$), with $p \approx 2 \times 10^{-3}$. Rank-$k$ CKA fails outright: because its feature map uses $|\lambda|$, it compares the kernels of $|A_c|$, discarding the eigenvalue signs in which class identity resides.

Table 12: QFS vs. linear CKA baselines on cross-dataset class pairs (seed 42, $k = 20$). CKA uses the union of MNIST and EMNIST-Letters test sets (CoM-normalized) as evaluation distribution; QFS and cosine similarity are computed from weights alone.

| Pair | QFS | $\mathrm{CKA}_{\text{rank-}k}$ | $\mathrm{CKA}_{\text{scalar}}$ | Cosine |
|---|---|---|---|---|
| 0–O (similar) | 0.437 | 0.801 | 0.680 | 0.698 |
| 1–I (similar) | 0.333 | 0.840 | 0.663 | 0.673 |
| 2–Z (similar) | 0.350 | 0.809 | 0.289 | 0.662 |
| 5–S (similar) | 0.455 | 0.809 | 0.506 | 0.679 |
| 0–X (control) | −0.093 | 0.811 | 0.515 | 0.679 |
| 1–W (control) | 0.153 | 0.657 | 0.000 | 0.646 |
| 3–H (control) | −0.218 | 0.639 | 0.094 | 0.655 |
| 7–O (control) | −0.109 | 0.762 | 0.040 | 0.668 |
| 4–A (control) | −0.144 | 0.768 | 0.022 | 0.671 |
| 6–F (control) | 0.008 | 0.792 | 0.022 | 0.682 |
| 8–L (control) | −0.050 | 0.763 | 0.007 | 0.667 |
| 9–R (control) | −0.012 | 0.732 | 0.013 | 0.630 |

# I  CP-Decomposition Results

Table 13: CP-Decomposition results (mean $\pm$ std, 5 seeds). Best CP accuracy-rank trade-off in bold. Accuracy throughout is on the MNIST 10k test split (labeled "validation" in training logs and figure axes). The Pareto figure additionally shows an $R = 784$ point ($784 = d_{\text{in}}$, the full input rank), extending the sweep of Section 6. Dense baselines are mini-batch trained ($\sim$2,900 optimizer steps); the budget-matched row and all CP variants are full-batch trained (100 steps; Section 6.4), so only same-protocol rows are directly comparable.

| Model | Accuracy (%) | Eff. Rank |
|---|---|---|
| Dense (no reg) | $97.49 \pm 0.06$ | $38.5 \pm 0.7$ |
| Dense (WD only) | $97.49 \pm 0.05$ | $15.5 \pm 0.3$ |
| Dense (WD, budget-matched) | $92.44 \pm 0.05$ | $6.1 \pm 0.2$ |
| Fixed $R = 256$ | $91.89 \pm 0.09$ | $3.74 \pm 0.07$ |
| Lambda $R = 256$ | $\mathbf{93.80 \pm 0.14}$ | $\mathbf{17.50 \pm 0.36}$ |
| Gated $R = 256$ | $93.79 \pm 0.15$ | $17.35 \pm 0.48$ |
| *Mini-Batch Protocol (Section 6.4)* | | |
| Lambda $R = 256$ (mini-batch, wd 0.1) | $97.81 \pm 0.07$ | $15.3 \pm 0.1$ |
| Lambda $R = 256$ (mini-batch, wd 1.0) | $97.72 \pm 0.03$ | $12.1 \pm 0.2$ |

Beyond the featured pair of Section 6.4 (dense weight-decay vs. Lambda CP), seed-averaged same-class QFS across all (dense config $\times$ CP mode) combinations at $R = 256$ lies between 0.07 (full regularization $\times$ Fixed CP) and 0.20 (no regularization $\times$ Lambda CP) with off-diagonal means near $-0.02$; the full grids and per-pair statistics are in the released results (`qfs_cp_bridge.json`). One implementation note: the Lambda variant's forward pass additionally applies a soft-threshold mask that zeroes components whose scale falls below 5% of the maximum; at convergence no component falls below this threshold in any $R = 256$ seed (verified in the released `qfs_bridge_robustness.json`), so the mask is inactive and the low effective ranks are attributable to the scale parameterization and weight decay.

## J  Full CP-Decomposition Results

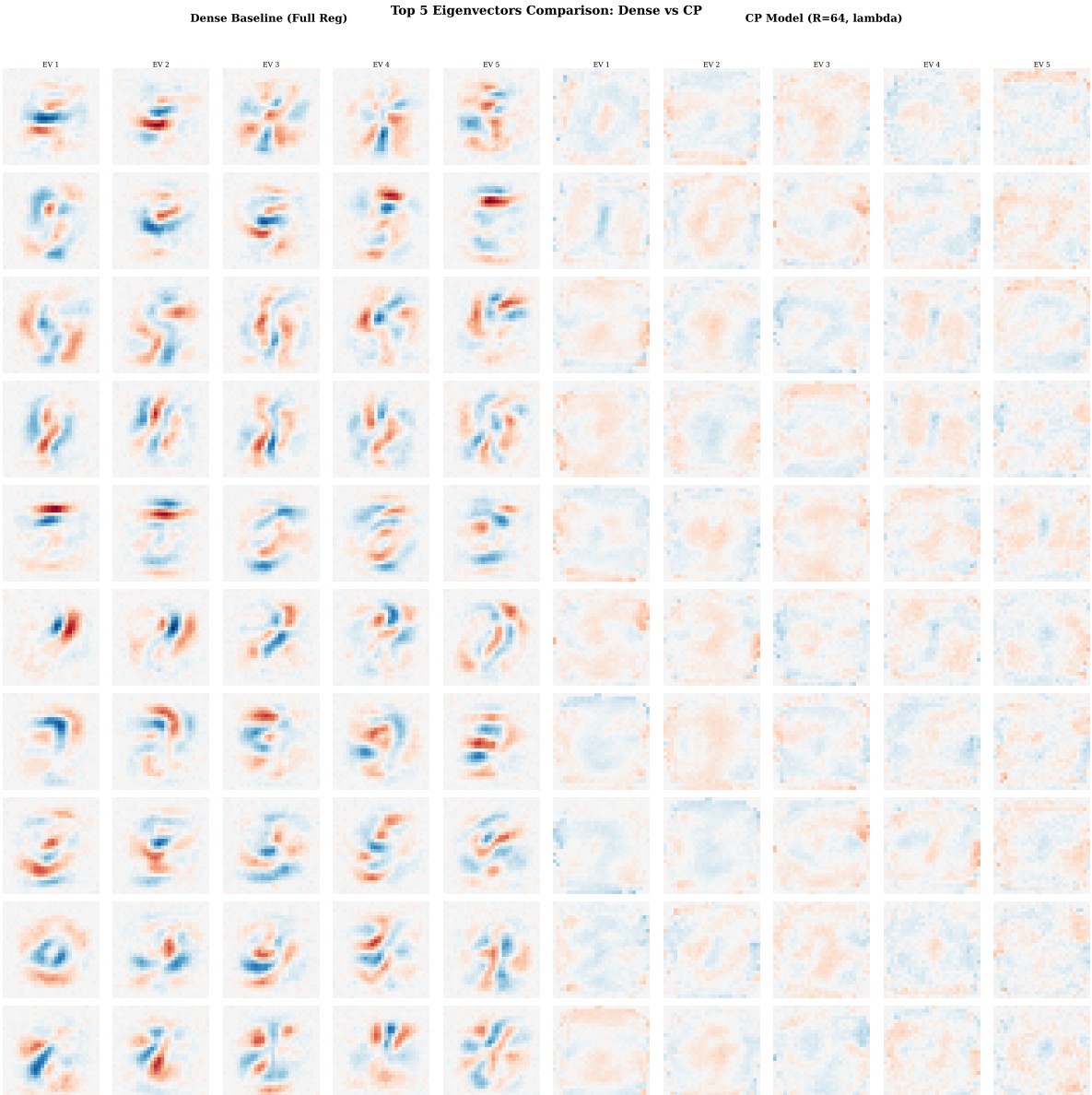

Figure 23: **Feature Disentanglement: Dense vs. CP.** Top 5 eigenvectors for digits 0–9. **Dense:** Holistic, superimposed templates mixing multiple strokes. **CP:** Atomic, localized factors (isolated curves, specific strokes) that can be combined to reconstruct digits.

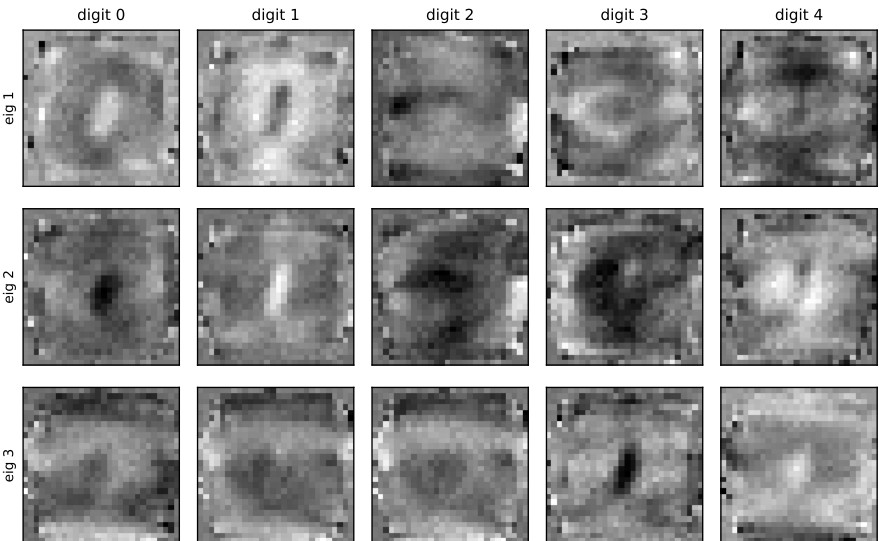

Figure 24: **Localization control.** Top-3 eigenvectors per class of a budget-matched dense model (full-batch, 100 optimizer steps; Section 6.4). They are diffuse and noisy rather than parts-like, so the localized CP factors of Figure 23 are not an artifact of CP's shorter training protocol.

Table 14: Complete CP-Decomposition rank sweep (mean ± std, 5 seeds).

| Model | Accuracy (%) | Eff. Rank |
|---|---|---|
| *Dense Baselines (mini-batch trained)* | | |
| Dense (no reg) | $97.49 \pm 0.06$ | $38.5 \pm 0.7$ |
| Dense (WD only) | $97.49 \pm 0.05$ | $15.5 \pm 0.3$ |
| *Budget-Matched Dense (full-batch, 100 steps; Section 6.4)* | | |
| Dense (WD, budget-matched) | $92.44 \pm 0.05$ | $6.1 \pm 0.2$ |
| *Fixed CP* | | |
| $R = 32$ | $90.63 \pm 0.23$ | $3.42 \pm 0.22$ |
| $R = 64$ | $91.05 \pm 0.13$ | $3.45 \pm 0.12$ |
| $R = 128$ | $91.48 \pm 0.22$ | $3.65 \pm 0.05$ |
| $R = 256$ | $91.89 \pm 0.09$ | $3.74 \pm 0.07$ |
| *Lambda CP* | | |
| $R = 32$ | $92.67 \pm 0.15$ | $9.07 \pm 0.44$ |
| $R = 64$ | $93.04 \pm 0.10$ | $11.57 \pm 0.52$ |
| $R = 128$ | $93.34 \pm 0.11$ | $14.92 \pm 0.49$ |
| $R = 256$ | $93.80 \pm 0.14$ | $17.50 \pm 0.36$ |
| *Gated CP* | | |
| $R = 32$ | $92.72 \pm 0.20$ | $8.76 \pm 0.47$ |
| $R = 64$ | $93.05 \pm 0.12$ | $11.69 \pm 0.35$ |
| $R = 128$ | $93.31 \pm 0.14$ | $14.83 \pm 0.16$ |
| $R = 256$ | $93.79 \pm 0.15$ | $17.35 \pm 0.48$ |
| *Lambda CP, Mini-Batch Protocol (Section 6.4)* | | |
| $R = 256$, wd 0.1 | $97.81 \pm 0.07$ | $15.3 \pm 0.1$ |
| $R = 256$, wd 1.0 | $97.72 \pm 0.03$ | $12.1 \pm 0.2$ |

## K   Tucker vs. CP: Identifiability on the Trained Tensor

To substantiate the identifiability argument of Section 6, we decompose the trained interaction tensor $B \in \mathbb{R}^{10 \times 784 \times 784}$ (config `full`, seed 42, symmetrized, float64) with both CP and Tucker at matched parameter

budgets (Table 15). Tucker fits better at equal parameters, as expected—its dense core is strictly more expressive—but the point of comparison is identifiability, not fit.

Table 15: CP vs. Tucker decompositions of the trained interaction tensor (config `full`, seed 42, the first of the five seeds) at matched parameter budgets; Tucker core size $10 \times 22 \times 22$. CP fitted by alternating least squares (ALS); its error is stable across three ALS restarts (0.536–0.539).

| Decomposition | Parameters | % of dense | Rel. Frobenius error |
|---|---|---|---|
| CP ($R = 25$) | 39,475 | 0.64% | 0.538 |
| Tucker $(10, 22, 22)$ | 39,436 | 0.64% | 0.354 |
| Dense tensor $B$ | 6,146,560 | 100% | — |

**Tucker is non-identifiable.** Rotating the first input-mode factor matrix by a random orthogonal matrix $Q$ ($U_2 \to U_2 Q$) while counter-rotating the core ($\mathcal{G} \to \mathcal{G} \times_2 Q^\top$) reproduces the tensor to machine precision (max absolute deviation $4.2 \times 10^{-17}$), yet the greedy-matched mean $|\cos|$ between original and rotated factor columns is only 0.46: the same tensor admits visually and numerically unrelated per-component "features" (Figure 25, top two rows). (Tucker's mode *subspaces* are essentially identifiable; it is the basis within each subspace—and hence any per-component feature—that is not.)

**CP is (approximately) identifiable.** CP-ALS restarted from three independent random initializations recovers largely consistent rank-1 factors up to permutation and sign: greedy-matched mean $|\cos| = 0.85$ on input-mode factors, with the best-matched components agreeing at $|\cos| \geq 0.98$ (Figure 25, bottom two rows). A minority of near-degenerate high-weight components remain initialization-dependent, reflecting that the trained tensor is only approximately low-rank (Kruskal's uniqueness condition (Kruskal, 1977) applies to exact decompositions). Per-component features are therefore meaningful objects under CP but basis-dependent artifacts under Tucker, motivating our architectural choice in Section 6.

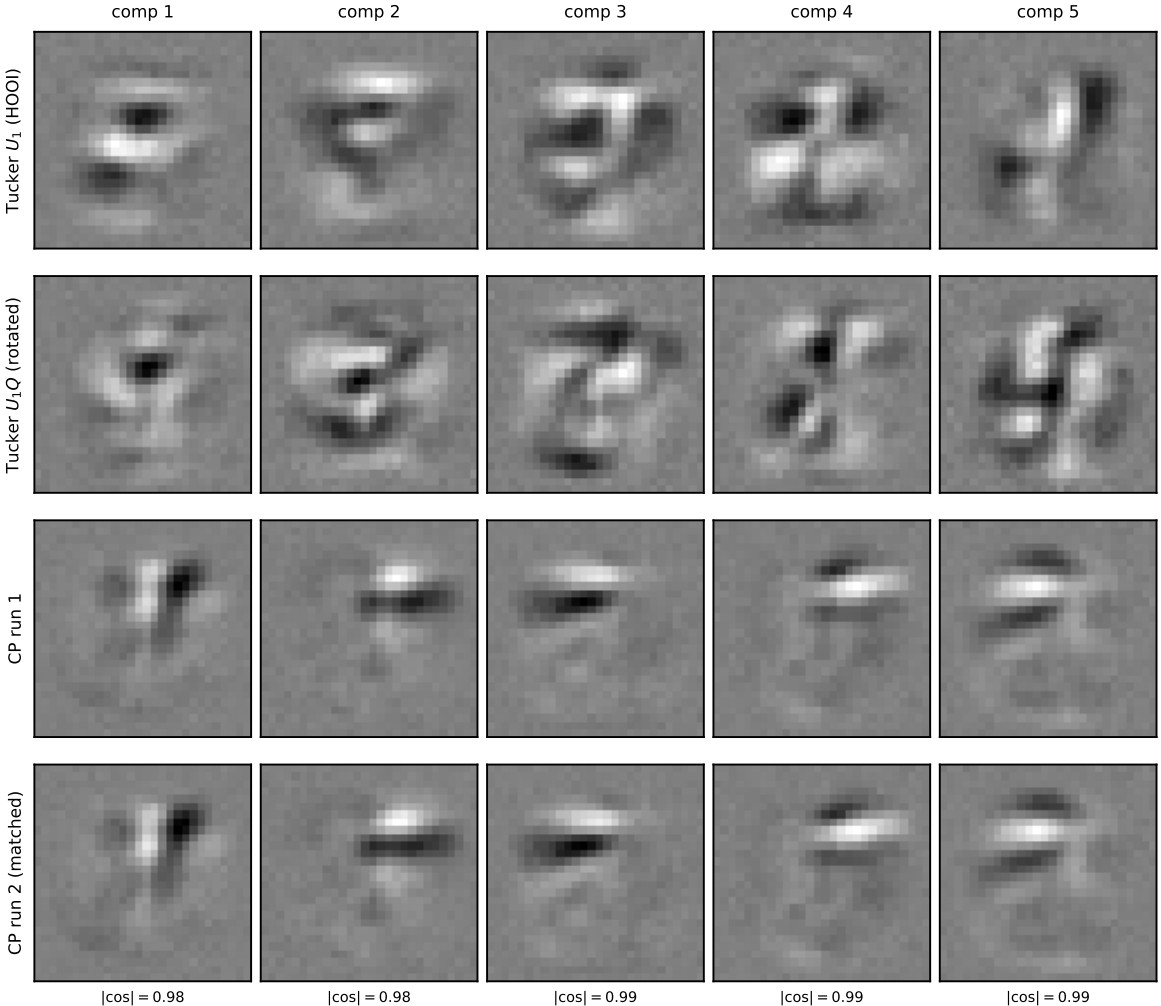

Figure 25: Identifiability of per-component features. **Rows 1–2:** leading Tucker input-mode factors (computed by higher-order orthogonal iteration, HOOI) before and after an orthogonal rotation absorbed into the core—the reconstructed tensor is numerically identical, but the "features" are unrelated. **Rows 3–4:** CP input-mode factors from two independent ALS initializations, greedily matched—the recovered rank-1 units largely coincide (per-panel $|\cos|$ annotated).

## L Environmental Impact

All experiments were tracked using `codecarbon` (machine mode) with Netherlands grid intensity. Hardware was mixed: vision training, cross-dataset, and CP experiments ran on an NVIDIA GPU node, while the language experiments and the challenge task ran on an Apple M4 (MPS, Apple's GPU backend), where some operations fall back to CPU. In Table 16, "GPU-h" denotes the wall time of runs executed on CUDA hardware (wall hours × number of CUDA devices); it is 0 for Apple Silicon runs, for which this proxy is unavailable. Wall times are not comparable across rows as measures of architectural efficiency: in particular, CP models train full-batch (100 optimizer steps per run) while dense models train mini-batch ($\sim$2,900 steps per run), which accounts for the small CP wall times. Total emissions: 0.169 kg CO2eq ($\approx$ 1 km driving).

Table 16: Environmental impact (CO2 via `codecarbon`).

| Experiment | Runs | Wall (h) | GPU-h | CO2 (kg) |
|---|---|---|---|---|
| Vision (Section 4)[†] | 101 | 1.66 | 1.11 | 0.040 |
| Language (Section 5)[‡] | 3 | 19.09 | 0.00 | 0.105 |
| Extension 1 (Cross-Dataset) | 15 | 0.58 | 0.58 | 0.020 |
| Extension 2 (CP-Decomposition) | 105 | 0.05 | 0.05 | 0.004 |
| Revision addendum (mini-batch CP, Sec. 6.4)[§] | 10 | 0.12 | 0 | — |
| **Total** | **224 + 10** | **21.39 + 0.12** | **1.74** | **0.169** |

[†]Includes 20 challenge task experiments (Figure 6), which ran on Apple Silicon and therefore contribute 0 to the GPU-h column.
[‡]Language experiments ran on Apple Silicon (MPS); GPU-h is 0 by the CUDA-wall-time definition above.
[§]CPU-only revision runs; CO2 not tracked (`codecarbon` unavailable in that environment).
Totals are computed before rounding, so columns may differ from the sum of displayed entries by one unit in the last digit.

