# OpenReview forum: "A Reproduction Study of Weight-Based Mechanistic Interpretability in Bilinear MLPs"
_TMLR — Under review for TMLR_

### Review · Reviewer_V7Cf · 2026-06-21

**Summary Of Contributions:**

The paper presents a reproduction study of Pearce et al. (2025), which introduced weight-based mechanistic interpretability via eigendecomposition of bilinear MLP interaction tensors. The vision reproduction is clean while the language is partial. Beyond reproduction, the paper introduces two extensions. One is the QFS, which is a similarity metric for comparing quadratic decision surfaces that outperforms cosine similarity on cross-dataset transfer tasks, and the other one is CP-decomposition, which is an architectural constraint that can achieve similar accuracy with training speedup.

Key Strengths:

1. The vision reproduction is thorough and clean, supported by multiple ablations.

Key Weaknesses:

1. The contribution is somewhat limited to reproducibility analysis.
2. The language reproduction is less directly comparable.

**Audience:**

Yes

**Audience Explanation:**

This paper addresses a timely topic of mechanistic interpretability and reproducibility. The practical finding that SAE convergence and model training compute materially affect interpretability metrics is useful for researchers designing or evaluating interpretability pipelines, and the study provides valuable perspective on the reliability of claims made about bilinear MLPs and SAE-based methods. That said, the audience remains relatively narrow, as the work is primarily scoped around reproducing a single prior paper, and the broader impact is somewhat limited.

**Broader Impact Concerns:**

There are no concerns regarding ethical implications, but it would be great to provide a more concrete discussion of the broader contribution to the mechanistic interpretability community beyond confirming and reproducing existing results.

**Claims And Evidence:**

Yes

**Claims Explanation:**

Strengths
1. *The vision reproduction is thorough and clean, supported by multiple ablations*: The vision claims are well-supported. The authors did multi-seed experiments, ablation tables, eigenspectrum visualizations, and truncation curves.


Weaknesses
1. *The contribution is somewhat limited to reproducibility analysis*: The extension claims about QFS and CP are supported only qualitatively and with limited experiments on a single bilinear layer and MNIST only. It would be great to compare with established alternatives such as CKA or Tucker decomposition. This could help make the contribution of QFS and CP more rigorous.
2. *The language reproduction is less directly comparable*: In the language reproduction part, the claim that the fraction of features exceeding a rank-2 correlation threshold is reported without confidence intervals. The language setup deviates from the original (8× rather than 4× SAE expansion for fw-medium and different layers per model for correlation analysis), which makes it less comparable.

**Requested Changes:**

1. The abstract and introduction list QFS and CP-decomposition as contributions, but these are evaluated only on MNIST with a single-layer model, without comparison to related alternatives. This makes these two contributions less rigorous, and it would be beneficial to examine them more systematically. It would also be great to provide comparisons with other established representation similarity metrics.

2. It would be better to provide more discussion on how the insights from this reproducibility study can be applied. It is unclear what the broader contribution is beyond confirming one paper's vision results and partially failing to reproduce its language results.

3. It would be great to provide confidence intervals in the language reproduction experiment.

---

### Review · Reviewer_pkhX · 2026-06-26

**Summary Of Contributions:**

This is a reproducibility paper on understanding role of and interpretation of MLP (Multi Layer Perceptron) component in a neural network-based prediction models. It reproduces two main experiments (one in vision domain and other in language domain) conducted in "Bilinear MLPs enable weight- based mechanistic interpretability" by Michael Pearce, Thomas Dooms, Alice Rigg, Jose Oramas, and Lee Sharkey. The paper proposes use of BiLinear MLP instead of standard non-linear activation function based MLP architecture. This results in interaction tensors that can be further decomposed (via eigendecomposition) to obtain interpretable features from the weights directly. This is advantageous compared to other techniques that require a separate model to be trained.

The author is able to reproduce the vision experiments and results match with the original paper, however, the language experiments are not fully reproduced. Further, the paper provides extension to existing vision experiments. In the first experiment, authors show that features learned by MLPs are structural in nature and can be transferred across vision tasks. In the second experiment, authors apply CP decomposition with the Bilinear architecture, this provides improved feature understanding and efficiency gains.

**Audience:**

Yes

**Audience Explanation:**

Yes, some of the researchers working in the mechanistic interpretability area would be interested in knowing how well can the results be reproduced and generalized.

**Broader Impact Concerns:**

The paper is a reproducibility paper and does not have any direct ethical concerns. It aims to advances in interpretability of LLMs and hence contribute towards general understanding of inner workings of LLMs.

**Claims And Evidence:**

Yes

**Claims Explanation:**

Yes, authors conduct experiments in the original paper. Authors are able to reproduce the original vision experiment results. In particular, the paper proves (empirically via experiments) that regularization induces low-rank structure in the bilinear interaction tensor. The paper also proves the claim that bilinear MLPs maintain near-full accuracy with low-rank truncation. However, for the language task, authors are not able to reproduce the results of the experiment on negation circuit discovery using pretrained bilinear transformers and SAEs. This is possibly due to public unavailability of 4x expansion SAEs. Consequently, the experiments are not exactly reproduced but show similar trends as the original paper.

Further, the paper extends the work by including CP (Canonical Polyadic) decomposition of BiLinear component within the architecture itself thus inducing intrinsic interpretability. As shown via experiments, CP decomposition results in more localized features and is more efficient in terms of scalability.

**Requested Changes:**

1. The paper is not well written, the author assumes that many of the details of the parent paper already known to the audience, which may not always be the case. Further, it doesn't explain many of the concepts clearly, it directly jumps into it. To someone who has not worked deeply in mechanistic interpretability, this paper is hard to understand.

2. Authors provide extension to the work by using CP-decomposition, however it is not clear what motivates this choice. While there are other possible tensor decomposition techniques, the paper does not give adequate explanation for using only CP decomposition.

3. As also state in limitations, extension experiments are conducted only on vision tasks but not on language task. It is not straight forward to extend these since it involves SAE, authors should explore this direction and conduct CP-decomposition experiments on language task as well.

---

### Review · Reviewer_yWna · 2026-07-01

**Summary Of Contributions:**

This paper reproduces the experimental results from [Pearce et al. (2025)](https://arxiv.org/abs/2410.08417), "Bilinear MLPs enable weight-based mechanistic interpretability". The original paper proposes that: the interaction between a bilienar MLP's transformation matrices across the output axis can be organized as a third-order tensor, which can be studied with standard linear algebraic techniques such as eigen decomposition or singular value decomposition. This reproducibility study revisits the original paper's experiments and finds that: 1) the original paper's claims on image classification models are largely reproducible, and 2) partially reproducible on language modeling tasks.

Aside from reproducing the experiments, this paper sharpens some of the claims/statements made in the original paper:
1) Confirms that weight decay are the main factor that drives the sparsity/clean-up of eigenvectors in bilinear MLPs.
2) Validates original paper's hypothesis/statement that: Low-rank approximation of output feature directions via top eigenvectors *drastically improve with longer SAE training runs*.

And, finally, this papeer also makes a couple of novel contributions:
1) In Section 5 proposes a new metric Quadratic Form Similarity, as cosine similarity (used in the original paper) fails to distinguish geometrically similar pairs.
2) In Section 6, using Canonical Polyadic (CP) decomposition during training to bake-in sparsity in the weights.

**Additional Comments:**

N/A

**Audience:**

No

**Audience Explanation:**

This is actually my main concern about this paper. This paper is a reproducibility study of a *single paper*, Pearce et al. (2025). If somebody wanted to learn about bilinear MLPs and their implications for mechanistic interpretability, I personally think they would be better off reading the original paper. In fact, I think the authors of this reproducibility study assumes that the reader is already familiar with the original paper. I had to keep the original paper open and frequently refer to it to understand many of the claims/design choices made in this reproducibility study.

The discussion in this paper is not opinionated enough, its kind of repetitive and a rephrasing of the original paper's discussions. Now, this study does make some technical contributions of its own (please see my summary of contributions above), which might be interesting to very niche readers aiming to pursue this specific line of research. But I personally think this contributions/insights are not significant enough to warrant a publication in TMLR. Also a personal opinion, I think this paper is much more suited for a workshop paper in a conference like NeurIPS or ICLR.

**Claims And Evidence:**

Yes

**Claims Explanation:**

Many of the experiment setups and design choices are directly borrowed from the original paper, and I think the authors have done a good job at this. Sometimes the authors specifically target one of the design choices to sharpen the claims made in the original paper: like confirming whether weight decay or noise augmentation contributes more to sparsifying the eigenvectors.

The presentation and figures in technical sections are clear and convincing.

**Requested Changes:**

As explained above, my main concern is that this paper is a reproducibility study of a single paper Pearce et al. (2025), and naturally, this paper heavily relies on that single paper. The specific research direction of weight-based interp with Bilinear MLPs is very niche with not that many papers. So, I am not really sure what the authors can do to make this paper more interesting to a broader audience.

Again a reiteration of what I said on the previous section, I didn't find the discussions novel, and the insights significant enough on their own (comparing with the original paper). Quadratic Form Similarity and CP decomposition are indeed novel contributions, but they are not discussed in depth on the main text. If you really believe on those, I would suggest the authors to expand on those contributions with more experiments/analysis and maybe frame the paper around those contributions, instead of framing it as a reproducibility study.

On other specific suggestions:
1. The Abstract needs a significant rewrite. Abstract is supposed to give the high level *abstract* idea/insights of the paper, and I think the current version utterly fails to do so. The current abstract has too many low level details and lists a bunch of numbers, that when read out-of-context of the main paper, simply doesn't make any sense.

2. Pearce et al. wasn't the first paper to propose *intrinsic interpretability through architectural design*. In fact, the object of study here, the bilinear MLPs for interp was discussed in [A technical note on bilinear layers for interpretability](https://arxiv.org/pdf/2305.03452) by Lee Sharkey, a co-author of Pearce et al. (2025). And, under that broad framing of interpretability through architectural design, many other papers also qualify as well: [Concept Bottleneck Models](https://arxiv.org/abs/2007.04612) by Koh et al. (2020), [GAMI-Net](https://arxiv.org/abs/2003.07132) by Yang et al. (2020), among others.